

# Exploring the 4D scales of eco-geomorphic interactions along a river corridor using repeat UAV Laser Scanning (UAV-LS), multispectral imagery, and a functional traits framework.

Christopher Tomsett[1], Julian Leyland[1]

[1]School of Geography and Environmental Science, University of Southampton, Highfield, Southampton, SO17 1BJ, UK.

*Correspondence to*: Christopher Tomsett (C.Tomsett@soton.ac.uk)

**Abstract.** Vegetation plays a critical role in the modulation of fluvial process and morphological evolution. However, adequately capturing the spatial variability and complexity of vegetation characteristics remains a challenge. Currently, most

of the research seeking to address these issues takes place at either the individual plant scale or via larger scale bulk classifications, with the former seeking to characterise vegetation-flow interactions and the latter identifying spatial variation in vegetation types. Herein, we devise a method which extracts functional vegetation traits using UAV laser scanning and multispectral imagery, and upscale these to reach scale guild classifications. Simultaneous monitoring of morphological change is undertaken to identify eco-geomorphic links between different guilds and the geomorphic response of the system in the

context of long-term decadal changes. Identification of four guilds from quantitative structural modelling based on analysis of terrestrial and UAV based laser scanning and two further guilds from image analysis was achieved. These were upscaled to reach-scale guild classifications with an overall accuracy of 80% and links to magnitudes of geomorphic activity explored. We show that different vegetation guilds have a role in influencing morphological change through the stabilisation of banks, but that limits on this influence are evident in the prior long-term analysis. This research reveals that remote sensing offers a

solution to the difficulty of scaling traits-based approaches for eco-geomorphic research, and that these methods may be applied to larger areas using airborne laser scanning and satellite imagery datasets.

## 1. Introduction

Fluvial eco-geomorphic interactions are co-dependent, complex, and variable across space and time, representing a continued area of interest within river research (Thoms and Parsons, 2002). The diversity of eco-geomorphology in river corridors can

be attributed to surrounding land use, existing morphology, and flood regimes (Naiman et al., 1993), whilst this same diversity simultaneously influences the flow of water and sediment, ultimately affecting morphology (Diehl et al., 2017) and floodplain conveyance (Nepf and Vivoni, 2000). The role of vegetation within the river corridor is well established, benefiting the local ecology (Harvey and Gooseff, 2015; Sweeney et al., 2004) alongside playing a role in natural flood management schemes and reconnecting channels and floodplains (Lane, 2017; Wilkinson et al., 2019), especially for small catchments where land cover

is more influential for flooding (Blöschl et al., 2007). This is important when considered against a backdrop of a rapidly





changing climate where flow extremes are more varied, flooding more likely (Unisdr and Cred, 2015), and riparian vegetation is likely to undergo shifts in composition (Rivaes et al., 2014; Palmer et al., 2009). Consequently, adequately measuring and monitoring vegetation with the fluvial domain is critical to understanding how these systems will respond to varying climatic and hydrological conditions.

The characterisation of riparian vegetation distribution over larger (>1 km) scales has typically relied upon the use of coarse classifications such as those identified in the Water Framework Directive (e.g. Gilvear et al., 2004), using techniques such are aerial imagery and satellite remote sensing (see Tomsett and Leyland, 2019). Any characterisation must be scalable and geographically transferable to cover the vast range of different fluvial landscapes whilst still accounting for the complexity presented within river corridors. Over-simplified, coarse classifications may altogether miss the vegetation complexity that exists, whilst conversely, highly detailed models tend to be necessarily localised and less transferable to alternate systems and scenarios. Traits-based classifications, developed and used within ecology, offer a scalable and transferable approach which can be applicable to the fluvial domain (Diehl et al., 2017). They have been shown to be useful for modelling topographic response to changing vegetation, sediment, and flow conditions (Diehl et al., 2018; Butterfield et al., 2020). However, challenges remain in broad application of this approach, with the characterisation of vegetation in the highly detailed manner required to extract traits metrics being challenging over larger (e.g. >1 km) scales.

To address these gaps, herein we examine the scales over which different traits can be collected from remote sensing methods and assess how well these traits can be used to establish eco-geomorphic relationships. We use a UK based temperate river as an exemplar site to demonstrate the effectiveness of novel remote sensing techniques for characterising vegetation. We investigate the limits of trait detection and the scales at which they are most appropriately used to enhance eco-geomorphic understanding, enabling us to establish the applicability of these methods to a variety of river corridor environments. Below we introduce the concepts of plant functional traits and hydraulically relevant traits before establishing the aims of this research.

### 1.1. The Importance of Vegetation

It is well understood that vegetation plays a key role within the river corridor and that how vegetation is modelled can affect the outcomes of hydrodynamic simulations. Channels with in-stream vegetation may experience roughness values an order of magnitude higher than non-vegetated channels (De Doncker et al., 2009), capable of reducing velocities by up to 90% (Sand-Jensen and Pedersen, 1999). However, foliage type and how vegetation is modelled affect the influence that the vegetation has on flow (James et al., 2008). The challenges posed by quantifying in-stream vegetation means that it is often difficult to make estimations of in-stream roughness (O'hare et al., 2011). Conversely, above water vegetation is easier to measure and monitor depending on the scales of analysis. Banks are typically eroded via mechanisms of mass failures or entrainment (Hughes, 2016) and so any stabilising effects of vegetation must influence these processes. Vegetation can reduce stream power, increase soil cohesion, and influence soil moisture levels, all of which can help to reduce erosion (Simon et al., 2000; Fox et al., 2007;



Kang, 2012). Bank collapse is influenced by three dominant factors, the extra mass of the vegetation, the shear strength provided by root reinforcement, and changes to bank pore water pressure (Wiel and Darby, 2007), with above ground biomass therefore directly influencing the mechanical and hydraulic properties of the substrate (Gurnell, 2014). The above ground biomass also has a direct influence on river flow and sediment transport when submerged (Gurnell, 2014), although this is stage dependent and depends on plant volume and structure.

## 1.2. Plant Functional Traits

Functional traits originate from ecological research, whereby criticism of using functional types led to a need for a more robust system of classification for ecological studies. Functional types represent vegetation based on its morphology and physiology, amongst other factors (Box, 1981; Box, 1996), but these attributes can exhibit greater variation within functional types as opposed to between them (Reich et al., 2007; Wright et al., 2005), as well as not varying between different types at all (Van Bodegom et al., 2012). Assessment of plants based on their functional traits has been seen as a method to overcome the shortcomings of the classic typological approach (Quétier et al., 2007).

Much like the attributes of a plant type, plant functional traits are morphological, physiological, or phenological attributes that are measurable at the individual plant level (Violle et al., 2007; Kattge et al., 2011; Savage et al., 2007). These measures can either be direct measures of a function such as photosynthesis or be a surrogate measure for a function such as leaf area. To be classed as 'functional' for ecology, traits must affect either plant growth, reproduction, or survival (Violle et al., 2007). Traits can either be effect or response based, depending on whether they have an influence on or are influenced by their wider environment (Violle et al., 2007). The benefit of traits-based methods is the applicability between different sites without needing species specific data (Mcgill et al., 2006). Therefore, the findings of community response to factors such as land use or climatic gradients (e.g. De Bello et al., 2006; Garnier et al., 2006) can be applied to a different location with similar trait composition. This is possible through the creation of guilds which describe plant groups with similar traits (Lytle et al., 2017) providing a scalable framework for eco-geomorphic research.

Traits-based approaches are well suited for eco-geomorphic research due to the strong environmental gradients within fluvial systems (Naiman et al., 2005). Vegetation responds to hydrological variables, such as water availability and disturbance events (Hupp and Osterkamp, 1996) whilst also influencing flow, sediment transport, and morphological stability (Gurnell, 2014), meaning that the bi-directional nature of this relationship maps well onto a traits-based framework. O'hare et al. (2016) have assessed the traits of nearly 500 species that influence river processes, revealing evidence of a broad link between plant form, distribution, and stream power within the UK (O'hare et al., 2011). Moreover, traits-based approaches allow for a more comprehensive view on eco-geomorphic interactions than a purely taxonomic approach due to the environmental conditions having a larger influence on trait compositions than species compositions (Göthe et al., 2017; Corenblit et al., 2015).



To date, the majority of traits-based research has focussed on ecological responses to hydrological conditions. For example, inundation likelihood has been shown to increase the presence of plants with longer and younger leaves (Stromberg and Merritt, 2016; Mccoy-Sulentic et al., 2017) whilst also being less woody (Kyle and Leishman, 2009; Stromberg and Merritt, 2016), with frequent inundation and higher stress environment necessitating greater flexibility. Conversely, plants in lower stress environments tend to be taller with longer life cycles (Kyle and Leishman, 2009; Stromberg and Merritt, 2016; Mccoy-Sulentic et al., 2017). Factors such as nutrient loading (Baattrup-Pedersen et al., 2016; Lukacs et al., 2019), light conditions (Baattrup-Pedersen et al., 2015), carbon availability (Lukacs et al., 2019), and anthropogenic interference (Baattrup-Pedersen et al., 2002; O'briain et al., 2017) are all key controllers of trait composition, with the environmental conditions better related to trait, rather than species, composition (Göthe et al., 2017). Furthermore, individual species have been shown to demonstrate differing traits depending on external stresses. *Populus nigra* trees were found to be smaller, have greater flexibility, and had a higher number of structural roots at a bar head when compared to a bar tail (Hortobágyi et al., 2017). Further work demonstrated that the smaller species at the bar head were incapable of trapping sediment when compared to those at the bar tail (Hortobágyi et al., 2018), highlighting the importance of traits rather than taxonomic approaches.

Hydrological variability can also influence trait assemblages. For example, mean flood frequency across 15 sites was found not to be related to trait diversity, whereas the magnitude of a 20-year flood and the variability in flood frequency were both related (Lawson et al., 2015). Controlled field experiments with artificial flooding and drought showed a decrease in species richness in both scenarios, although trait diversity was more tolerant to drought conditions (Baattrup-Pedersen et al., 2018). Rivers with more variable flows tend to encourage pioneer species and those with prolonged drought seeing an increased abundance of water tolerant species (Aguiar et al., 2018). As a result, these responses mean successful river restoration projects should focus on the type of restoration more than the extent (Göthe et al., 2016). Taxonomic approaches can still perform equally well for fluvial studies, but traits-based approaches tend to account for local and regional conditions better (Tabacchi et al., 2019), which is necessary for scalability.

Research into effect traits and their geomorphic influence has received less attention as traits concepts have only recently started to be explored in hydrological research. However, as noted by Corenblit et al. (2015), the interactions between plant traits and fluvial systems are linked, with hydrological conditions affecting plant establishment and survival and plant morphological traits affecting morphology and subsequent establishment. There is evidence that changing traits can alter the morphological evolution of channels, with invasive species that have higher branching densities and less flexibility increasing aggradation through reductions in near bed velocities (Manners et al., 2015). Guild location impacts the morphological response, with analysis of bars showing different responses downstream and also laterally based on the traits of the dominant species in these directions (Hortobágyi et al., 2018). This is supported by Butterfield et al. (2020) who when examining changes in multi-annual elevation found that guilds at different locations, experiencing different hydraulic conditions, had differing impacts, but also that guilds could not explain all the variation in morphological response. It was found that differing canopy



architectures that interacted with flow were likely to be the prominent driver of topographic response, supporting the research of Manners et al. (2015). However, trait diversity can impact morphological response as much as the individual traits, with combinations of guilds interacting to alter responses (Hortobágyi et al., 2018), from which spatially averaging to areas of dominant guilds may oversimplify the complexity of interactions.

### 1.3. Hydrologically Relevant Functional Traits

Not all vegetation functional traits are relevant when considering direct relationships between vegetation, hydrology, and morphology. Moreover, not all traits can be obtained from remote sensing techniques, a necessary requirement when upscaling to larger domains. Below we identify the vegetation traits that are directly relevant to river systems and which can potentially be captured via remote sensing techniques, thereby allowing the upscaling of any developed methods of characterisation.

Existing studies that have considered vegetation-flow interactions have focused on plant height and frontal area as key metrics which explain momentum exchanges in river flows. The height of the plant affects the amount of interaction (Nepf and Vivoni, 2000), with varying flow depth determining the proportion of the plant frontal area which is submerged. Frontal area is an often used proxy for the scale of obstruction and is a component of the drag formulation which can have a larger impact on flow conditions than the selection of a drag coefficient (Järvelä, 2004; Wilson et al., 2006). However, the limitations of 2D metrics to describe the complex nature of plants has been highlighted, with the use of 3D data and plant volume offered as improved methods (Whittaker et al., 2013; Vasilopoulos, 2017).

Under various flow conditions, the frontal area of a plant may change due to flexing and reshaping, with studies showing that not accounting for this can limit the results of drag models (Sand-Jensen, 2008; Whittaker et al., 2013). A higher leaf area increases the momentum absorbing area of plants with de-leafed vegetation not bending until a higher threshold velocity is reached (Wilson et al., 2003; Järvelä, 2002a). Drag has been calculated using leaf area, although not a 1:1 relationship it was shown to be suitable for estimating vegetative resistance (Jalonen et al., 2012). The contribution of foliage to resistance decreases with flow speed, Whittaker et al. (2013) noting a drop in the drag contribution of foliage from 75% to 20-50% at speeds under and over 0.5 ms$^{-1}$ respectively. This is due to the reshaping of plant structure during higher flows leading to reductions in drag (Armanini et al., 2005), with the rate at which this reduction happens being plant dependent (Järvelä, 2002b; James et al., 2008; Boothroyd et al., 2017). The vertical distribution of plants also has a significant impact on flow, with different vertical distributions such as step changes or continuous variations, impacting flow differently and being more important than multi-plant arrangement (Lightbody and Nepf, 2006; Jalonen et al., 2012).

The arrangement of plants is still important in determining bulk drag, with drag coefficient values for a single foliated stem not representative of stems occurring in bulk vegetation (James et al., 2008). Higher plant densities within a channel lead to an increase in drag coefficients, however the arrangement of vegetation within the channel has a negligible impact (Järvelä,





2002b; Kim and Stoesser, 2011). Sand-Jensen (2008) identified that there was a difference in downstream flow between evenly distributed plants and the same biomass distributed into high density clumps, with the former providing the larger increase in drag and impeding flow the most. Therefore, spatial variation in plant distribution may be more important than the density of the patches themselves. A higher stem density does result in more scour around stems and deposition to be further from the scour sites, however overall deposition does not increase with increased stem density (Follett and Nepf, 2012).

Whilst both vegetation structure and distribution of individual plants directly impact flow, many other vegetation traits can impact sediment transport processes, for example through playing a role in altering the erodibility of periodically submerged banks or bar surfaces, or through increased resistance from root structures. Although vegetation height, frontal area, and leaf area are all key effect traits which can be measured directly, accounting for secondary impacts of vegetation related to below ground biomass for example, and how all traits vary spatially and temporally remains the challenge for advancing our understanding of eco-geomorphic interactions.

### 1.4. Remote Sensing of River Corridor Vegetation

Although many of these traits are inherently measurable in the field, many of them are not obtainable from current remote sensing methods. Direct trait extraction for riparian vegetation from airborne (i.e. large scale) remote sensing has not yet been utilised to enhance eco-geomorphic studies. Currently, collection of trait data relies on direct ground based field surveys and lab analysis, or species are identified in the field and traits taken from databases (e.g. TRY database (Kattge et al., 2020)). Methods are often dependent on site access, species richness, and variation within the study area (Palmquist et al., 2019), utilising methods such as quadrat surveying or transect sampling. This technique is effective for establishing traits but is limited by the spatial extent of ground coverage. Some variables inevitably require databases to avoid substantial disturbance, such as root characteristics (e.g. Stromberg and Merritt, 2016; Aguiar et al., 2018; Baattrup-Pedersen et al., 2018), although databases should be used with caution; for example, maximum plant height is not related to the plant submergence height at the time of a particular flow event, and great variation can be seen in both effect and response traits for a singular species (Hortobágyi et al., 2017; Hortobágyi et al., 2018). Therefore, accounting for temporal and spatial variation in traits is important and highlights the need for temporally and spatially relevant data collection.

For fluvial research, multispectral imagery can be used to determine species using supervised and unsupervised classifications with good accuracy (Butterfield et al., 2020). Outside of fluvial research there is an increasing awareness of the potential of remote sensing methods to help drive the scalability of functional traits, especially in relation to physical traits such as plant height, leaf area index, phenology, and biomass (Abelleira Martínez et al., 2016; Aguirre-Gutiérrez et al., 2021), yet considerable limitations remain due to the uncertainty in relating spectral and physical properties to functional traits (Houborg et al., 2015). Upscaling localised high resolution data is possible however, for example from TLS (Terrestrial Laser Scanning) to large scale ALS (Airborne Laser Scanning) data (Manners et al. (2013).



Advances in UAV (Uncrewed Aerial Vehicle) remote sensing can create an important link between these two scales of data collection. UAV data collection allows high resolution imagery and active remote sensing methods such as laser scanning to 200 be conducted on large reaches relatively easily (Tomsett and Leyland, 2019), increasing coverage and providing a middle ground for relating local to large scale data. Multispectral cameras have already helped to improve the classification of vegetation from UAVs (Al-Ali et al., 2020), and active UAV-LS (UAV Laser Scanning) has also been shown to be comparable in estimating tree structures to TLS methods (Brede et al., 2019). Such methods therefore present an opportunity to not only classify vegetation by types and assign them to guilds, but to define guilds based on characteristics acquired from remote 205 sensing directly, before upscaling this to reach scale classifications.

### 1.5. Aims

The aim of this research is to develop a set of scalable traits-based 3D vegetation metrics which can be used to assess spatial and temporal (i.e. 4D) variation and importance of eco-geomorphic interactions on an exemplar UK river system. This is achieved using the following specific objectives:

1. Undertake an assessment of the longer term (multi-decadal) eco-geomorphic evolution of the channel using satellite remote sensing, to compare planform evolution within vegetated and non-vegetated channel sections.
      2. Identify and select hydrologically relevant traits which can be extracted from high resolution remote sensing data.
      3. Establish the presence of vegetation guilds (those with similar traits) for the river reach, based on exploratory analysis and object orientated random forest classifications.
4. Compare the spatial extent of these guilds to morphological change over the study period to establish eco-geomorphic feedbacks.

### 2. Study Site

The exemplar site is located on the upper course of the River Teme on the English-Welsh border in the UK (Figure 1A). The study area consists of two broader regions; the upstream section consisting of open grassland with patches of heterogeneous 220 vegetation, and the downstream section which flows through denser vegetation. The River Teme is a highly mobile, gravel bed river within an alluvial floodplain which exhibits numerous avulsions. There is active lateral erosion of the channel, depositional gravel bar features, and woody debris dams across the study site (Figure 1A). The reach has typically low flows (Figure 1B), with an average depth of 0.69 m (+/- 0.15 m) throughout the year with slightly higher average flow depths in the winter months (November – February, 0.79 m +/- 0.15 m). 95% of river depth has been below 0.99 m and 99.9% of the flow 225 depth has been below 1.48 m. The largest recorded river depth was 2.85 m on the 16th February 2020 during Strom Dennis. Figures are obtained from a gauge station 3 km downstream of the study site, starting from the earliest gauge record.

Earth **Surface**
**Dynamics**
Discussions
EGU

**Figure 1 Study Site of the River Teme and the long term water level at the Knighton gauge station 3km downstream. A) Study Site**
**Location on the River Teme, UK. Inset images show active bank erosion and a large debris dam caused by falling trees.**
**Orthoimagery collected February 2020 and background imagery provided by ESRI (2021). B) River Gauge Level at the Knighton**
**monitoring station ~1 km downstream from study reach (data available from 2002 – present).**

## 3. Methods

### 3.1. Long Term (Decadal) Analysis

To assess the longer-term context of eco-geomorphic interactions within the study reach, historical satellite imagery was
analysed to identify channel mobility in relation to riparian vegetation. Channel mobility was assessed by digitising bank edges
across multiple years. This method is well established and has been used previously to study the evolution of a large river
confluence (Dixon et al., 2018) and for multi-decadal analysis of a single river (Yao et al., 2013; Gupta et al., 2013), to identify
the drivers of morphological change. These have typically been restricted to coarse (e.g. 30 m ground resolution) satellite
datasets, with planform change only detectable if it is greater in magnitude than the image resolution. This can result in mixed
pixels; where multiple land cover and vegetation types are misidentified into one category (Henshaw et al., 2013). Here we



make use of high spatial resolution imagery from Google Earth (0.5 - 2 m, source dependent (Google Earth Pro, 2021)) and Pleiades (0.5 m) to identify historical changes in channel location and vegetation cover. Google Earth historical imagery for the years 2000, 2006, 2008, and 2009 and Pleiades data from 2013, 2015, 2016, 2018, and 2020 were used from which bank

lines were digitised, resulting in 20 years of channel evolution. Banks under tree cover were identified where possible using a mix of spectral bands (Pleiades data only) to highlight channel position. To account for the images being taken at various time periods throughout the years and subsequently having different flow regimes, bank tops were digitised as opposed to water edges to reduce uncertainty resulting from variable flow stage. The exception to this was where no clear bank top was present, for example on the large bars, where evidence of usual high flows from colour changes and trash lines in the imagery were

used to guide digitisation of bankfull channel width.

All analysis of bank movement was performed in ArcGIS using the Digital Shoreline Analysis System (DSAS, (Himmelstoss et al., 2018)) with a 1.5 km long baseline created for both left and right banks based on the dominant river planform trend. Transects were cast every 5 m and manually edited where necessary in order to intersect the outermost bank, especially on

tight meander bends. The Shoreline Change Envelope (SCE), the distance between the nearest and furthest bank from the baseline, is used to infer total channel mobility throughout the reach.

To assess the impact of vegetation, the channel was classified into two classes: those containing structurally large vegetation and those that did not. Areas classed as containing structurally large vegetation could either include a small number of trees

clumped around the channel, a linear section of vegetation on one bank, or larger areas of vegetation such as woodland. These regions were user defined based on all of the image sets available and were used to group transects within regions containing large vegetation and those that did not, for comparison of the SCE statistics. As vegetation may have an influence on both the local scale and broader reach scale morphology, the analysis was repeated for changes excluding the reoccupation of new or former channels (classed as avulsions). To achieve this, DSAS transects that spanned across two separate channels from

different years were removed. Each individual channel was then reanalysed using separate baselines, consequently the impact on the results from channel switching can be isolated and removed.

Statistical comparison was undertaken of the SCE values for sections containing large vegetation and sections that did not. These could be used to identify any differences in the SCE values and therefore inferred mobility of these sections, and the

influence vegetation may have on planform evolution. To investigate the morphological process of avulsions, the development of new channels between satellite images was also tracked. New and developing channels which were visible in satellite imagery were digitised for each set of images. These were compared to UAV flood extent imagery from February 2020 alongside historical LiDAR data from 2007 of the river corridor and qualitatively assessed in relation to how topography and flood events influence planform, and the processes by which channel switching occurs.





**3.2. Field Collection of High Resolution 4D data**

High resolution UAV-LS (UAV Laser Scanning) and UAV-MS (UAV Multispectral Imagery) were collected over the entire reach through February 2020 until June 2021, capturing all seasonality. To complement these flights, Terrestrial Laser Scanning (TLS) surveys of vegetated and bar sections were undertaken to gain a benchmark ultra-high-resolution dataset for comparison to the UAV-LS and for characterising small herbaceous vegetation. A UAV-RGB (Red Green Blue) survey was
also undertaken during overbank flow on the falling limb of Storm Dennis in February 2020, to identify the flood extent. Table 1 summarises the survey dates, extents, and data collection methods.

**Table 1 Data collection methods, extent and point density for each survey date. TLS point density is based on the resultant point cloud after registration. Ground Sampling Distance (GSD) is the resolution of the resultant orthomosaics. UAV-LS point density is**
**taken once cleaning of the raw clouds has taken place.**

| Date | Survey | Sensor | Point Density/GSD |
|---|---|---|---|
| 06/02/2020 (Winter) | Whole Reach | UAV-LS | 778 m$^{-2}$ |
| | | UAV-MS | 0.04 m GSD |
| 18/02/2020 (Winter) | Whole Reach | UAV-RGB | 0.02 m GSD |
| 16/07/2020 (Summer) | Subsection | UAV-LS | 810 m$^{-2}$ |
| | | UAV-MS | 0.04 m GSD |
| | | TLS | 16,000 m$^{-2}$ |
| 14/09/2020 (Autumn) | Whole Reach | UAV-LS | 762 m$^{-2}$ |
| | | UAV-MS | 0.04 m GSD |
| 14/04/2021 (Spring) | Whole Reach | UAV-LS | 791 m$^{-2}$ |
| | | UAV-MS | 0.04 m GSD |
| 03/06/2021 (Summer) | Whole Reach | UAV-LS | 804 m$^{-2}$ |
| | | UAV-MS | 0.04 m GSD |

A detailed outline of the UAV based sensor set up, processing routine and accuracy assessment can be found in Tomsett and Leyland (2021), with a short overview of the system provided below. UAV-LS and UAV-MS were collected using a DJI Matrice 600 Pro multirotor aircraft, capable of flying for 20 minutes per flight. Two sets of batteries allow for the spatially
complex 1 km reach of the River Teme to be captured with some redundancy. Multispectral imagery was obtained from a MicaSense RedEdge MX camera, collecting imagery with a ground resolution of ~0.035 m across five spectral bands, consisting of blue (475 nm), green (560 nm), red (668 nm), red-edge (717 nm), and near infra-red (842 nm) wavelengths (Micasense, 2021). The laser scanner is a Velodyne VLP-16 Puck Lite, firing 16 laser-detector pairs at approximately 300,000





points per second, with a 360° horizontal and 30° vertical field of view. The sensor has a range of up to 100 m and a typical
ranging accuracy of +/- 0.03 m (Velodyne Lidar, 2016). Both sensors use direct georeferencing from an Applanix APX-15,
which utilises multi-frequency GNSS and MEMS (Micro Electro-Mechanical System) inertial motion unit to provide post
processed positional and orientation accuracies up to 0.02 m and 0.025° respectively (Applanix, 2016). This removes the need
for extensive GCP placement throughout the reach. Georeferenced point clouds from the laser scanner and Structure from
Motion based point clouds and orthomosaics from the multispectral imagery were produced, both with vertical accuracy under
0.1 m. UAV-RGB imagery was collected from a DJI Inspire 2 with a Zenmuse X4S camera, resulting in a ground resolution
of 0.017 m from a flight height of 60 m. An on-board EMLID REACH M2 provides positioning accuracy of up to 0.015 m
when post-processed (Emlid, 2021), with a connection to the on board camera to allow image captures to be timestamped to
assist with the SfM processing. TLS data was captured in July 2020 using a Leica P20 Scanstation, collecting high resolution
(0.0031 m point spacing at 10 m distance from scanner, resulting in a mean point density of 16,000 points per $m^2$ within the
area of interest) scans of two locations. The first, an area of channel containing large vegetation at the inlet of the study site
(two convergent TLS scans), and the second, part of a large meander bend in the centre of the study area (four convergent TLS
scans) where large vegetation was absent. Targets were used to register scans together, acquired using a Lecia TS06 total
station, with a resultant scan registration accuracy of +/- 0.007 m.

### 3.3. Vegetation Functional Trait Extraction

The workflow developed to extract plant functional traits consisted of five steps: (1) Separation of individual plant point clouds
that could be used for analysis, (2) Analysis of these individual clouds to extract metrics related to their traits, (3) Separation
of plants into guilds adapted from Diehl et al. (2017) based on similar traits, (4) Identification of guild properties extractable
from temporal UAV-LS and UAV-MS datasets for reach scale classification inputs, and (5) Use of an object-based random
forest classifier to determine the spatial discretisation of guilds.

#### 3.3.1. Point Cloud Segmentation

A number of automatic methods exist to classify very dense point cloud scenes into different groups (e.g. Brodu and Lague,
2012; Zhong et al., 2016). However, the majority of these are designed for very high-resolution TLS datasets and so here a
semi-automated approach was employed. Smaller vegetation whose structural composition cannot be fully resolved from
UAV-LS data were analysed from the summer TLS survey. Automatic classification of ground/non-ground points was
performed using the progressive morphological filter in the LidR package (Roussel et al., 2020) before manually segmenting
in CloudCompare (https://www.danielgm.net/cc/) to create individual plant models (Figure 2, *Raw Point Cloud*).

For the herbaceous plants, leaves and flowering parts were removed from the clouds so as not to interfere with the quantitative
structural modelling (QSM). Although foliage is important, for the methods used herein they could not be accounted for due



to point densities. Any statistical outliers were detected, removing points 2.5 standard deviations and above the mean distance between points, resulting in a dataset consisting of 37 herbaceous plants.

Tree segmentation also used a combination of manual and automatic classification based on surveys in leaf-off conditions exposing the full internal tree structure. 24 trees were selected from across the reach representing a range of structures and

sizes from which complete models could be created. As above, initial separation of ground and vegetation points was performed using a progressive morphological filter. Whilst automatic classification methods such as CANUPO exist (Brodu and Lague, 2012), the UAV-LS point densities necessitated the manual extraction of individual trees, prior to interactive filtering using a number of statistical measures. Local volume density helped to separate points distinct from the main tree woody structure, whilst linearity metric filters (how aligned points are within a set radius) remove points that are highly

complex or not part of the main tree structure. The statistical outlier removal tool and a final manual check can then be used to remove any remaining erroneous points. This resulted in a point cloud of predominantly large branches, with a clearer structural profile as can be seen in Figure 2 (*Filtered Point Cloud*). The thresholds for separating individual trees are size, structure, and point density dependent, hence the need for interactive selection.

This adds an element of user bias as to what is deemed a 'main' branch, but the lower density of UAV-LS scans makes this a necessary method before reconstructing vegetation models (Brede et al., 2019). Shrubs and grasses whose structure could not be fully resolved from the UAV-LS or TLS data were not analysed for traits extraction. Aside from requiring many TLS scans to capture the extensive and complex branching networks of these plants, in eco-geomorphic terms a traits-based rather than bulk roughness approach is likely to be limited.

**3.3.2.    Trait Metric Extraction**

For the reconstruction of vegetation stems into cylindrical models, the open source TreeQSM method was applied to the partitioned UAV-LS and TLS derived vegetation data (Brede et al., 2019). TreeQSM utilises 'patches' to determine connected points in the vegetation cloud, before growing the tree structure by joining patches together to form a complete model (Raumonen et al., 2013). These are created using user defined initial patch sizes to adjoin points, before refining the patch

sizes using minimum and maximum sizes to create a complete model. This allows the coarse branch structure of the tree to be identified (Figure 2, *Segmented Point Cloud*). Sections are then generalised as cylinders for computational efficiency and as they provide a robust representation of trees (Raumonen et al., 2013). These cylinders can then be used to describe the overall structure and properties of the individual plant (Figure 2, *QSM Cylinder Model*). A full method description can be found in Raumonen et al. (2013). QSM methods have been noted to overstate the volume of smaller branches and are sensitive to noise

in the data alongside variable point density (Fang and Strimbu, 2019; Hackenberg et al., 2015). However, QSM reconstructs tree structures in a manner which resolve many of the hydraulically relevant vegetation traits.





**Figure 2 Vegetation trait extraction, from an individual raw point cloud to a cylindrical model and frontal area. The process is demonstrated for two extracted vegetation point clouds, a large tree within the study reach collected from UAV-LS data, and a small perennial on the central bar collected from TLS, note the difference in scales. The segmented point cloud is coloured by branching order from blue to red, with the cylinders coloured in the same manner.**

Patch diameters (which are used to determine adjacent points within the same tree) were chosen following a parameter sensitivity exercise, with the range of values initially based around those of Raumonen et al. (2013) and Brede et al. (2019) for TLS and UAV-LS approaches respectively. A visual assessment was performed to identify parameters that created models similar to the point cloud structure due to the lack of reference data. After testing for the optimum patch sizes for reconstruction, the TLS scans of herbaceous vegetation initial patch diameter was set at a size of 0.005 m, with the second patch diameter





minimum and maximum sizes of 0.002 and 0.01 m. The minimum cylinder radius was set to 0.005 m, prescribing the smallest detectable branch structure of the extracted herbaceous plants. For the UAV-LS derived tree data, the initial patch diameter

was 0.2 m, with the second patch dimeter minimum and maximum sizes of 0.1 and 0.5 m. The minimum cylinder radius was 0.1 m, based on manual measurements of tree branches within the point cloud that were detectable. For each plant model the cylinder reconstruction and variable extraction was repeated ten times. As the modelling begins at a random location each time the start point can affect the results, and so multiple averaged simulations provides a more accurate solution. The modelling produces a number of metrics, but for this study height, number of branches, diameter at breast height, volume, and maximum

branching order, were collected. For each metric of interest, the average value and standard deviation of these values are taken from the ten runs.

The frontal areas of all segregated vegetation clouds were extracted alongside the construction of the cylinder models, based on the 2D methods described by Vasilopoulos (2017). For each discretised plant point cloud, the data was flattened from 3D

to 2D by collapsing the data along a single horizontal dimension on a regular grid (Figure 2, *2D Frontal Area*). The grid resolution was set at half the width of the minimal detectable feature resolved by the QSM modelling; 0.005 m for the TLS derived herbaceous plants and UAV-LS 0.05 m for UAV-LS derived trees.

### 3.3.3   Guild Identification

Based on the separated points clouds, each were assigned to a guild based loosely on hydrologically relevant traits outlined in

O'hare et al. (2016) and Diehl et al. (2017). As outlined above, a decision was made to discretise grasses and shrubs using bulk roughness metrics due to their relative homogeneity and the need for ultra-high-resolution data. Short branching herbs and taller single stemmed herbs were identified, with discrepancies in flexibility, branching, and height, likely to influence hydrology differently. Woody vegetation was further split in to two guilds, those with high diameter at breast heights (DBH) that had low density of trunks and those with lower DBH that had a higher trunk density. The analysis was preformed separately

for woody and herbaceous vegetation. As the aim was to identify characteristics that would separate out the guilds from remotely sensed data, there was little need to compare woody and herbaceous species directly as height would be a dominant component.

In order to assess whether remotely sensed data could separate out plants into their guilds in a statistically robust way, a

Principal Components Analysis (PCA) was undertaken to identify the variables which explained most variation within their derived metrics. The metrics used for the PCA analysis were those obtained from the QSM and frontal area calculations which were normalised to remove the influence of different scales (Alaibakhsh et al., 2017). The principal components identified were used to inform the classification of reach scale guilds, identifying those variables that most explained the variation between guilds.




### 3.2.4. Linking Traits to Reach Scale Metrics

To scale the analysis from individual plants to the entire reach level, a method of linking plant scale traits to broader scale data is required. Convex hulls representing the spatial extent for each vegetation point cloud extracted and analysed above were used to define the regions from which UAV-LS and UAV-MS data were extracted. For small herbaceous vegetation, this was buffered by 0.25 m to account for any misalignment between TLS and UAV-LS clouds. For tree vegetation polygons this buffer was increased to 1 m to incorporate peripheral branches and leaves removed during point cloud filtering. Polygons for small branching trees and large shrubs were created based on field observations and UAV-MS imagery. A total of 11 polygons were created for this combined guild category, with 11 made for grasses, 8 for water classes, and 5 for gravel bars and bare earth. Within these polygons, multiple seasonal variables were extracted for scaling local guild identification to reach scale classification. The structural characteristics of the point cloud were extracted through TopCAT (Brasington et al., 2012), obtaining the standard deviation, skewness, and kurtosis over a decimated grid at both 1 and 4 m resolutions, the latter to account for larger vegetation footprints. The 4 m resolution decimated grid only considered points classified as vegetation in the initial 'ground/other' point clouds to remove ground points from further analysis. To extract a Canopy Height Model (CHM), a bare earth digital terrain model (1 m resolution) was subtracted from a 0.25 m digital surface model incorporating the vegetation points. The Normalised Difference Vegetation Index (NDVI) across the reach was calculated using the red band along with both the red-edge and near infrared bands of the MicaSense orthomosaic images to produce two separate NDVI layers. As the red-edge can be used to separate out vegetation signatures, using a combination of both was expected to help differentiate plants with similar structural but different spectral properties. Analysis of structural and spectral data was performed for each of the four seasons to gain an insight in to how these properties vary temporally. For each of the vegetation polygons, the attributes of each of these layers for each season were extracted using zonal statistics. The mean and standard deviation for each attribute for each season were then calculated across the different guilds for use in the classification model.

### 3.2.5. Reach Scale Guild Classification

To scale from guilds created from individual UAV-LS and TLS derived plants, to the entire reach, an object based random forest classification was undertaken. Object based approaches overcome some of the issues of variation and complexity in high resolution images (Myint et al., 2011), improving continuity in the results (Duro et al., 2012; Wang et al., 2018). The RGB bands from the multispectral camera and the CHM were combined to create a 4-layer image from which to classify distinct objects. The Felzenszwalb Algorithm was applied which uses graph based image analysis to segment an image into its component parts based on the pixel properties (Felzenszwalb and Huttenlocher, 2004). This results in regions within the image being grouped base on them having similar properties according to the input layers, avoiding the salt and pepper effect found in traditional pixel by pixel classification approaches (Wang et al., 2018).





**Table 2 Description of guilds used for training the random forest classifier, showing the number of training objects from the image segmentation and the training area size.**

| Guild | No. of Training Objects | Training Area Size (m²) |
|---|---|---|
| Grasses | 93 | 321 |
| Branching Herbs | 15 | 25 |
| Single Stemmed Herbs | 16 | 29 |
| Branching Shrubs | 135 | 388 |
| Low DBH Trees | 158 | 876 |
| High DBH Trees | 62 | 238 |
| Bars | 122 | 641 |
| Water | 41 | 157 |


In total, 644 training objects were identified using the previously discretised vegetation convex hull regions, with multiple objects present within each training sample (Table 2). A random forest classifier was then trained based on the layers that were deemed to distinguish between the different guilds, having proved an effective machine learning technique (Adelabu and Dube, 2015; Chan and Paelinckx, 2008; Adam and Mutanga, 2009), with a water mask included to reduce errors associated with

varying flow stage. An analysis of model accuracy vs number of forests showed a convergence of accuracy above 100 forests and a reduction in band importance variability above 300 forests (Figure 3). Higher variation in band importance suggested that the number of trees was influencing the likelihood of an optimal solution. This random forest classification was then applied to the remaining objects within the reach.

Due to the limited number of samples being used, there were not enough training samples to split into a training and test dataset. The multi-tree approach of random forests is constructed on a sample of the dataset and as such can be tested against itself to determine an out of bag accuracy score. It also successively adds and removes bands to determine the band importance in the classification (Adelabu and Dube, 2015). Alongside this self-assessment, for the final guilds classes a total of 80 random points were generated across the study site with an equal number in each outputted guild. These were manually classified using

high resolution ortho-imagery from a UAV-RGB (0.02 m resolution) survey and study site knowledge. The output classification was not visible when undertaking this assessment and the order of the points shuffled to remove user bias. The classified guilds map was then used to extract the predicted guilds of these points before a confusion matrix was utilised to assess the accuracy of the classification.





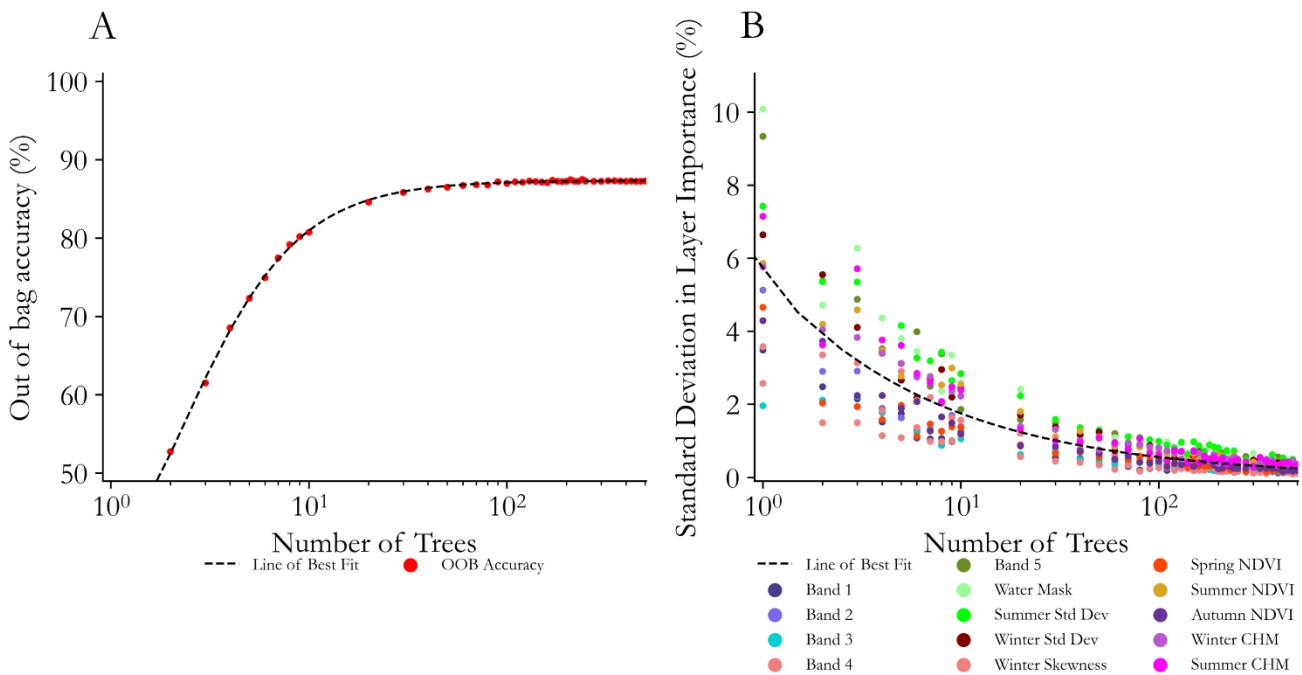

**Figure 3 Random forest classifier out of bag accuracy and variations in band importance for guild classification. A) Out of bag accuracy scores for different numbers of trees used within the random forest classification, showing a distinct levelling off in accuracy after ~100 trees are used. B) The standard deviation in individual band importance across 10 sample runs to identify at what number of trees band importance becomes consistent across all runs, in this instance around 300 trees.**

## 3.3. Morphological Change

The M3C2 algorithm (Lague et al., 2013) was employed to calculate morphological change, whereby the surface normals from a subsampled cloud of core points (here at 0.1 m resolution) are calculated, and change along the normal direction is identified with the calculation of a local confidence interval. This overcomes some of the limitations of traditional elevation model differencing which can't account for the direction of change. The benefits of using both SfM and UAV-LS data allows their respective drawbacks to be overcome through combining both datasets. SfM has been shown to perform poorly in vegetated reaches, whereas UAV-LS maintains good ground point densities, whereas SfM provides good continuity and high point densities in unobstructed areas. Therefore, in order to obtain good surface normals for assessing change the two clouds were merged (see Tomsett and Leyland (2021) for error analysis) and their vegetation removed through the use of the same progressive morphological filter used previously to produce resultant clouds which were then differenced using M3C2.



# 4.    Results

## 4.1. Decadal Scale Change

Analysis of planform shift from the year 2000 through to 2020 has identified that the channel is highly mobile, experiencing rapid change in places as well as more gradual evolution in others. From the 586 bank transects cast, the average SCE (extent of bank movement) was 38 m whilst the median change was 25 m. The smallest change was 1 m whereas the largest was 120

m. Comparison of sections with large vegetation present and absent suggests there is a greater average mobility in vegetation present sections (Table 3). This goes against the assumption that vegetation helps to reduce channel mobility. However, Figure 4 suggests that the areas where the channel has remained predominantly stable through time have some vegetation influence. Of the four areas of significant change, only one appears to follow the traditional meander development model of lateral erosion leading to a cut off, with the three remaining sections showing likely avulsion or previous channel reoccupation. Analysis of

channel mobility excluding these avulsions indicates that reaches with large vegetation present have lower rates of lateral mobility, and that there is evidence of large vegetation reducing rates of planform shift.

**Table 3 Transect statistics showing the difference between sections with large vegetation present and those where it is absent s, with and without channel reoccupations (avulsions). N refers to the number of transects within each category.**

| | SCE statistics for each scenario | | | | |
|---|---|---|---|---|---|
| | N | Mean | Median | Std. Deviation | Max. |
| Large Vegetation Present (Inc. Channel Reoccupation) | 220 | 48 | 24 | 41 | 121 |
| Large Vegetation Absent (Inc. Channel Reoccupation) | 348 | 35 | 27 | 27 | 101 |
| Large Vegetation Present (Exc. Channel Reoccupation) | 290 | 10 | 8 | 7 | 47 |
| Large Vegetation Absent (Exc. Channel Reoccupation) | 339 | 22 | 16 | 18 | 72 |


Figure 4 (A) and (B) compares frequency of SCE values for sections with and without large vegetation being present, including and excluding avulsions. For reaches with large vegetation, removing the avulsions has a notable impact on the distribution, with many more transects falling within smaller SCE values. Although this shift is seen in the transects without large vegetation also, the change in distributions is less prominent and is supported by a smaller change in mean values, dropping from 35 m

to 27 m for large vegetation absent reaches and from 48 m to 10 m for reaches with large vegetation present.



**Figure 4 Results from the decadal DSAS analysis. A) DSAS results showing channel evolution from 2000 – 2020, with left- and right-hand banks digitised for each year. Spatial variability in maximum planform shift is shown in blue. Background imagery provided by ESRI (2021). B) shows the range of SCE values from both the left- and right-hand banks combined, for sections classed as containing large vegetation and those that do not. C) shows the SCE values when the effects of avulsion are removed from the data, for both large vegetation present and absent sections also.**

This channel switching appears to involve reoccupation of former channels that have lower floodplain elevations during overbank flow events. The three erosion channels in Figure 5 show the stages of progression. Feature C demonstrates a completed channel neck cut off for a double meander bend. Both features A and B appear to be developing during flood events, with the orthomosaic insets showing the overbank flow captured during a flood event and the resulting channel position post



flood for feature B. This implies a consistent pattern in new channel development that occurs during successive overbank flow events.


**Figure 5 Historical analysis of avulsion development across the floodplain. The top panel shows two developing channels (A and B) across the floodplain and one channel neck cut off (C). The bottom left panel (Channel A) shows how this development is influencing successive planform shift, and the bottom right panels (Channel B) demonstrates how this is linked to overbank flow.**






## 4.2. Hydrologically Relevant Trait Analysis

### 4.2.1. Extraction and Analysis of Traits

The QSM analysis appears to output visually sensible results and produce models appropriate for the vegetation being modelling (see Figure 2). Table 4 shows the standard deviation of a selection of QSM metrics as a percentage of the mean

value. The repeat modelling was more consistent for larger vegetation, with lower relative standard deviations. However, for some metrics such as number of branches, herbaceous plants with few branches may be adversely affecting the results. For example, plants with 5 stems having errors of +/- 1 branch is a 20% difference, whereas for 20 stems this is only 5%. Overall, model repeats appear to have good agreement with one another, and provide a basis for separating out vegetation with similar hydraulic functional traits.


**Table 4 Standard deviations in trait values as a percentage of the mean values for herbaceous and tree guilds. Guilds aggregated to include all herbaceous and tree data. Expressed as a percentage of mean due to the varying scales of data between the two guilds.**

|  | **Height** | **Number of Branches** | **DBH** | **Volume** | **MBO** |
|---|---|---|---|---|---|
| Herbaceous Guilds | 3.87 | 16.77 | 17.83 | 12.18 | 17.52 |
| Tree Guilds | 1.16 | 8.79 | 15.58 | 12.89 | 15.00 |

Figure 6 shows the PCA plots of herbaceous vegetation metrics from the TLS scans (A) and woody vegetation metrics from the UAV-LS scans (B). It is clear that some separation of points through dominant metrics is possible, with both plots exhibiting two principal components capable of separating the defined guilds. Panel A shows the PCA plot for herbaceous vegetation. Height is a clear component between each guild, as well as volume. Although the number of branches was not a key component for separating guilds, branches per unit height explained some of the variability in the data. Taller plants may

have a similar number of branches, and so accounting for plant height produces a density of branches independent of size to help explain plant structure. Of the four identified components, only the height is identifiable from the UAV-LS data for upscaling, however, point density and spectral properties may improve guild separation. Panel B shows the PCA plot for woody vegetation. Height is less important in distinguishing the two guilds than for herbaceous vegetation, yet trees under or over certain heights are likely to be one guild or the other suggesting minimum and maximum threshold values. For separating

guilds, the most important components appear to be DBH and vertical skew which was expected as this was the basis for initial guild classes. DBH cannot always be easily extracted from UAV-LS data if it is incomplete, therefore as the vertical distribution acts in the same component direction, this can be used as a potential method for differentiating guilds. There is however considerable overlap in both of these PCA plots for woody and herbaceous vegetation. There are dominant trends such as the DBH and plant height for separation, but there is considerable variation within the guilds for their QSM based

metrics which may impact the final classification.



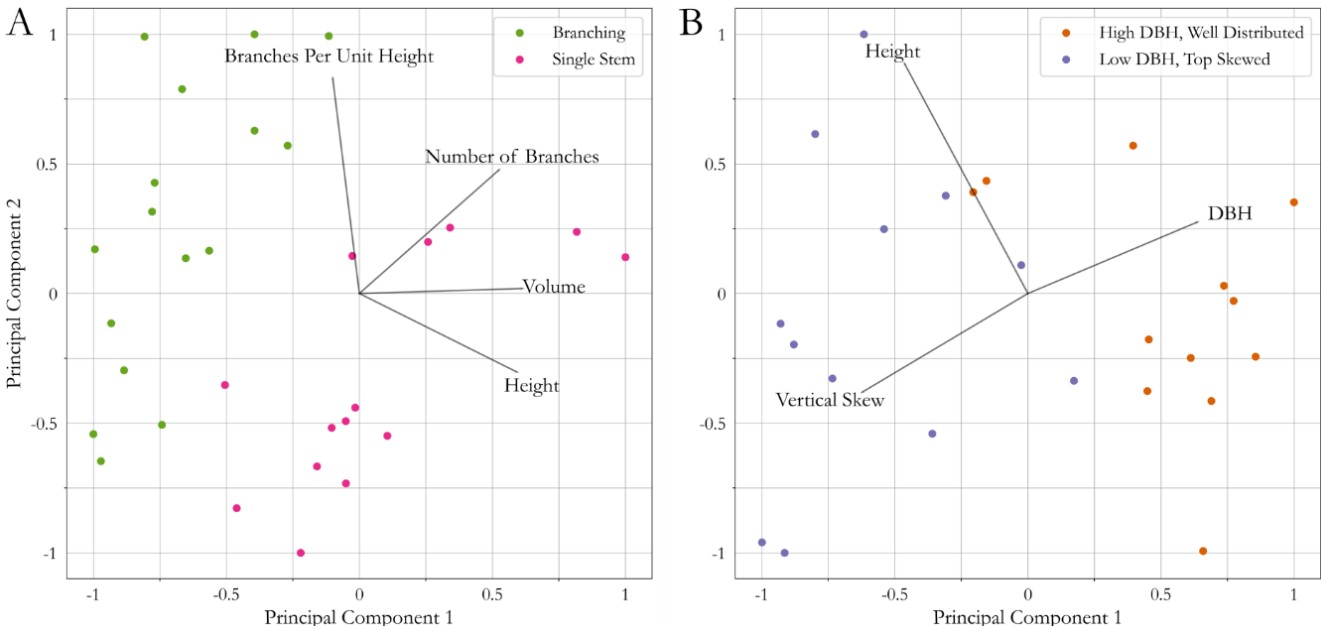

**Figure 6 PCA analysis of (A) herbaceous and (B) tree guilds to investigate differences in trait characteristics. Lines indicate direction of each variable that explains variation in the data.**

### 545    4.2.2.    Linking PCA Clusters to Reach Scale UAV-LS Data

Figure 7 shows the results of the seasonal analysis of different variables derived from UAV-LS and UAV-MS imagery for each of the guild classes. There are clear variables which can separate different guilds with ease, for example the height of the canopy is a key indicator between woody, herbaceous, shrub, and grass guilds. Separating out similar guilds does appear to be more nuanced. The High DBH and Low DBH woody guilds both have very similar values and seasonal patterns of changes in

NDVI values as well as in their height. This is unsurprising as the PCA analysis showed, with height not a dominant factor in explaining variation with numerous samples showing crossover. Vertical skew did show guild separation, with the samples used for QSM analysis collected in leaf off conditions. Figure 7 does suggest that changes in winter skew are visible between the two guilds, with a smaller amount of crossover as expected. Spring, summer, and autumn skewness is less informative, likely due to leaf on conditions effecting full tree reconstruction, with higher variability in results between the sample areas.


Separating out herbaceous guilds is also a challenge. Elevation values for single stemmed herbs are more variable and cross over in to grasses and multi-branching herbs. However, the mean elevation values are higher in line with the PCA analysis and may enable herbaceous guild separation. Likewise, the average skew values help to differentiate between classes, but again the variability in the data suggests it is harder to separate by structural content alone. Conversely, spectral data shows great





promise in differentiating between guilds. Both the absolute values between herbaceous guilds show different as well as their

seasonal patterns especially when utilising the red edge band for NDVI calculations.

**Figure 7 Results of seasonal analysis (X-axis within subplots) of different reach scale metrics (Y-axis) from UAV-LS and UAV-MS data for each identified guild. The point clouds at the top provide an example point cloud of each guild class, with canopy height**
**ranges acquired from trait extraction for the four analysed guilds and from the reach scale analysis for the remaining grass and shrub guilds. Error bars indicate one standard deviation around the mean, CHM (Canopy Height Model) is given in metres, IR refers to Infra-Red and RE to Red-Edge bands in the NDVI calculations.**





### 4.2.3. Creation of Seasonal Reach Scale Guilds Maps

The resultant classification from guild classification can be shown in Figure 8 with many areas being classified as expected.

There appears to be an over classification of the branching shrubs class based on initial comparisons with ortho-imagery, whereby the edges of larger vegetation and some predominantly grass regions appear to have been misclassified. This may be due to the large variation in structural and spectral characteristics of this guild which were less well accounted for. Herbaceous guilds were predicted in areas that were expected, in mobile areas of the channel were larger vegetation would find it more challenging to establish. The out-of-bag accuracy score when training the random forest classifier with 300 trees was 87.2%.

Figure 9 A shows the importance of each band in the classifier, with structural elements proving key in separating guilds, especially using summer standard deviation of point elevations. The near infra-red band and winter standard deviation are the next most important elements, with the remaining individual spectral bands providing a smaller contribution to the classification. The higher importance of the two NDVI layers implies that providing the classifier with analysed image data is more useful than individual bands alone. Likewise, the canopy models alone are less informative than the variation in

elevations when detecting guilds, supporting the use of manipulated rather than simple metrics to help improve classification.

The confusion matrix can be seen in Figure 9 B comparing the number of check points that are correctly and incorrectly predicted. The overall model accuracy is 80%, lower than the out-of-bag prediction. However, this is not surprising as training areas were delineated based on complete structural profiles for the QSM analysis and the total number of samples used for

training was small relative to the possible variation across the reach. There was a general over classification of points as grass guild, with only one grass control point incorrectly classed as branching herbs. Branching herbs which are more detectable from imagery and likely to return more laser scan points were classified reasonably well, only being misclassified as grass.

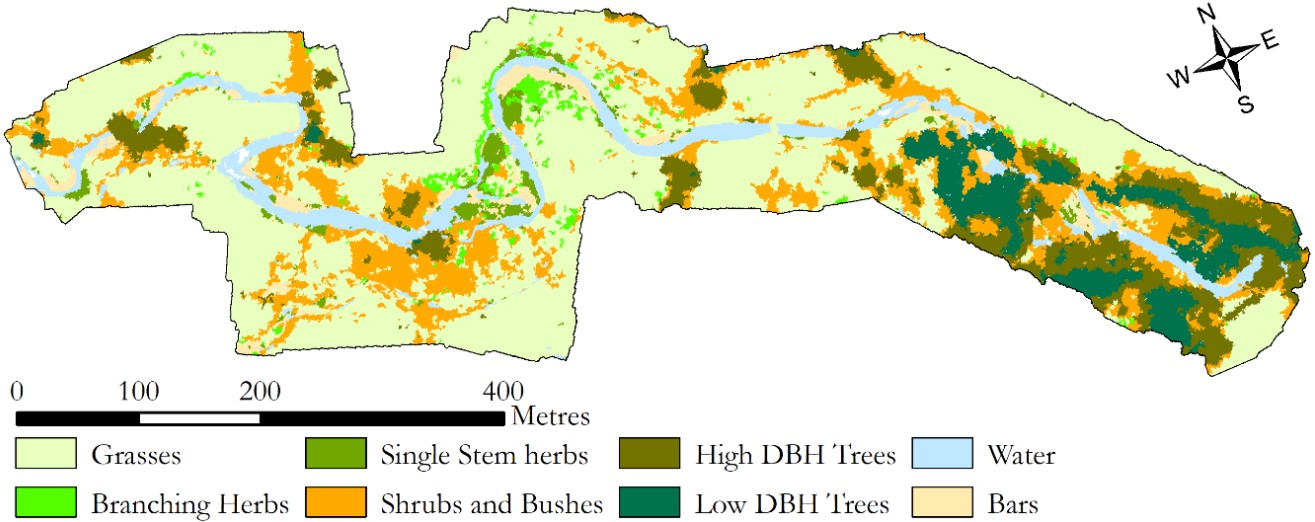

**Figure 8 Resulting classification from reach scale analysis for the areas covered by both UAV-LS and UAV-MS data. Note the over**
**classification of shrubs and bushes, especially at the edge of larger wooded guilds.**



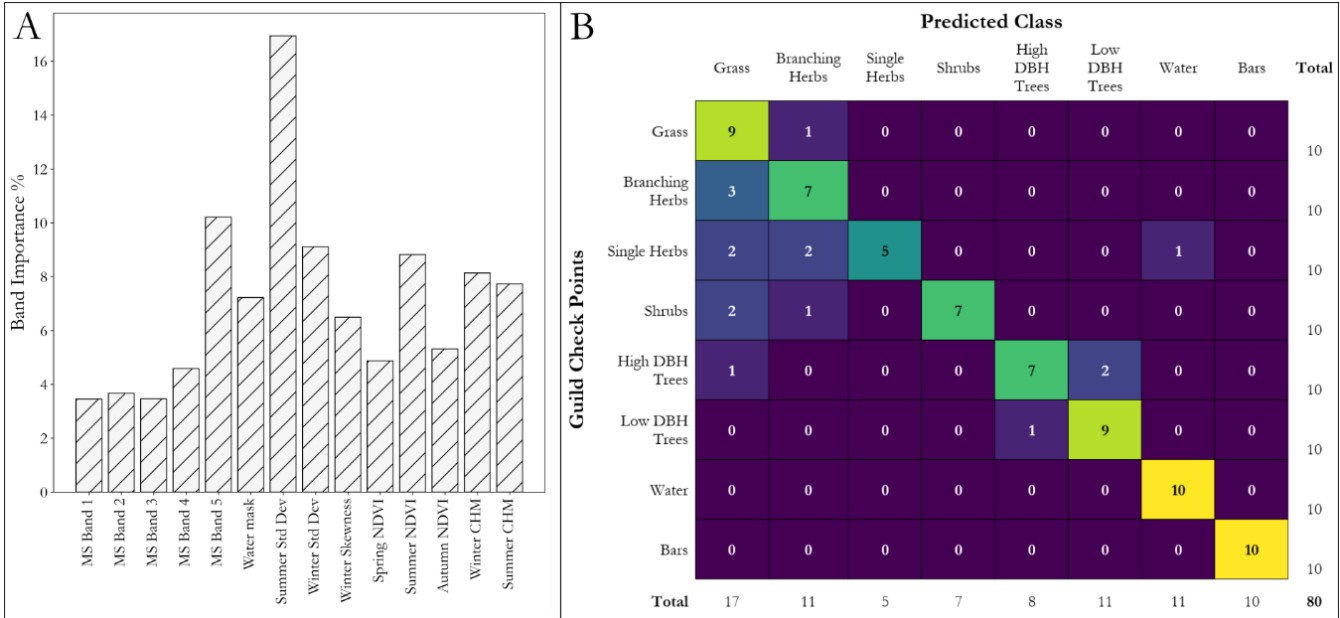

**Figure 9 Individual band importance in the final classification (A) and confusion matrix (B) from the accuracy assessment. The band importance represents the contribution of an individual layer to the final classification. The confusion matrix demonstrates for which guilds the classification struggled, showing an over-classification of grasses and the poor detection of single stem herbs. The overall classification accuracy was 80%.**

Single branching herbs however were relatively poorly classified (50% accuracy), being misclassified as grass, branching herbs, and even water. However, their narrow structure and sparse spacing make them hard to identify from coarser imagery and they return fewer laser scan points. This class also exhibited the greatest variation in values when using reach scale metrics to evaluate guild samples. Shrubs were predominantly misclassified as branching herbs and grass, this may be due to the object segmentation not always isolating complete plants or including surrounding ground points which may have affected the classification. Low DBH trees with a top skew were classified well by the model, most likely due to their larger heights and winter skew, whereas higher DBH trees were misclassified as both low DBH trees and grass. The former likely due to the difficulty in separating out these two guilds which have subtle differences in certain classification layers such as winter skew, and the latter from surrounding data being included in an object likely from shadowing continuing an object outside its true bounds. However, of all 20 tree check points, only one was incorrectly classified as a guild with clearly different traits, a High DBH Tree as Grass (see Figure 9).

### 4.3. Morphological Change

As is expected, the majority of morphological change occurs over winter months when there are high flows (Figure 10). Conversely, over periods of lower flow during the summer both the extent and magnitude of change is reduced. Throughout the first winter period erosion occurs on the outer bank edges with fairly consistent planform evolution throughout the reach. Deposition is evident throughout the entire reach, however erosion is considerably more dominant than deposition, with almost





14,000 m$^3$ of net erosion. The second winter appears to have more localised effects on morphology, with clear channel reshaping through the upper half of the study area. This has led to considerable deposition on both sides of the channel in areas of previously active erosion as well as localised erosional hot spots (~23,000 m$^3$ net erosion). Both histograms within the

winter seasons show a dominance in erosion overall. This is in line with previous long-term analysis which shows this as an area of high mobility with previous channel reshaping occurring. Over both winters, morphological change in the tree dominated downstream reach has undergone similar levels of change with areas or erosion and deposition influenced by the

**Figure 10 Morphological change throughout the monitoring period, showing the spatial variation in erosion and deposition as well as the net change in sediment. Note that February 20 – July 2020 is a composite DEM of difference consisting of comparisons between February and July to the left of the dashed line and February to September to the right of it. In July, only half of the survey area was captured. The stability of the reach over summer (July to September) justifies attributing change to the February – July result. The histograms adjacent to each time period show the distribution of magnitude of change, and whether this tends to be favouring net erosion or deposition.**





presence of large vegetation. Both summer periods have a greater degree of stability, with erosion and deposition taking place but in lower magnitudes. This is consistent throughout the reach with no hotspot areas of either deposition or erosion, with deposition showing to be more dominant overall.

## 5. DISCUSSION

### 5.1. Multi-Decadal Evolution

The multi-decadal evolution for this reach is complex and analysis of the formation of new channels implies that flood events might be a key control on the switching from one channel to another and the reoccupation of former channels. It is not possible to isolate a single variable that may cause such switches to take place, such as particular flow thresholds, baseline conditions,
vegetation, or soil characteristics. However, it does appear that areas influenced by large vegetation experience less localised bank evolution, with the vegetation constraining the channel to some degree. This does not appear to stop large switches in channel position into or away from vegetated sections. This implies that vegetation is playing a role in the stabilisation of channels up to some, as yet unidentifiable, threshold. The reoccupation of former channels implies that vegetation plays a lesser role than topography in these conditions, suggesting that whilst vegetation can have controls on channel evolution, these
eco-geomorphic feedbacks are locale and flow condition dependent. This supports the concept of vegetation acting as river system engineers and providing an influence on channel morphology (Gurnell, 2014) and that varying vegetation densities may be impacting the resistance to morphological evolution (Bertoldi et al., 2011). Therefore, at a decadal scale, although vegetation may not be the sole control on planform evolution, it is shown to be an important factor in this reach of the River Teme.

### 5.2. Trait Extraction and Guild Formation

Current measurements of plant functional traits are still predominantly ground based and therefore limited by on site access (Palmquist et al., 2019), requiring extensive sampling to extract enough data to create guilds relevant to a particular study (e.g. Diehl et al., 2017; Hortobágyi et al., 2017; Stromberg and Merritt, 2016). Remote sensing of these traits is therefore a potentially novel way to collect data across large areas, depending on the vegetation size and methods of data collection.
Although no ground truth data relating to traits was collected in the field, the assessment of variability in model construction suggests that the final cylindrical models were of good fit for the point clouds collected. This suggests that the use of remote sensing to collect structural trait data has an important role to play in eco-geomorphic research moving forward, especially once trade-offs in terms of time and spatial extent are accounted for.

The use of pre-determined rather than site specific guilds was a method employed by Butterfield et al. (2020) on the basis of guilds outlined in Diehl et al. (2017). The sites used in both of these studies were similar, and the application to a temperate





UK based site is challenging. However, the comparatively smaller sample size used in this study, and the lack of a comprehensive guilds list for riparian vegetation, made using predetermined guilds described in Diehl et al. (2017) and O'hare et al. (2016) justified. The lack of suitable ultra-high-resolution data reduced the number of herbaceous guilds to two, on which

most distinction could be observed. The variation in woody vegetation created two guilds within this single previously outlined guild, as they were likely to have different hydraulic effect traits. This basis appears to have proved effective with differences in structural characteristics which are likely to impact flow and subsequent morphology noted between the guilds during PCA analysis. Predominantly single stemmed herbs were taller, likely due to their improved structural standing, and although the number of branches was similar the number of branches per unit height was different. A taller, stronger, and less branching

herb is going to have a distinctly different impact than a shorter more flexible one (Nepf and Vivoni, 2000; Järvelä, 2004; Sand-Jensen, 2008). Being able to differentiate successfully between these two highlights the success of the survey and trait extraction methods. Likewise, the difference in flow conditions between low DBH trees that are closely packed to less densely packed high DBH trees may show a resemblance to the influence found at smaller scales on plant density (Järvelä, 2002b; Kim and Stoesser, 2011). The relationship between DBH and vertical skew is not surprising; considering the higher plant spacing

density the competition for space is likely higher, resulting in more mass higher up the tree profile. As plants cannot yet be easily differentiated by their DBH, using vertical skew gives promising results for upscaling to larger areas whereby ALS surveys may be able to differentiate between woody guilds for better informed hydrological analysis.

However, UAV-LS has been shown to overestimate canopy reconstruction volume (Brede et al., 2019), which mirrors the over

complexity demonstrated in Figure 2 (*QSM Cylinder Model)* with some awkwardly orientated cylinders. Extracting traits using remote sensing is novel and can outcompete ground-based methods for coverage but is not yet likely to match the accuracy and interpretive ability of in-field measurements. Moreover, use of TLS is highly localised with a limit to the survey extent that can be captured (Lague, 2020), meaning only a small number of samples can be analysed which may not reflect the full variation in vegetation morphology from differing hydrological and environmental conditions. The UAV-LS data, although

covering more ground, does take significant levels of time to post-process and extract multiple individual vegetation models, although as the spatial extent of coverage increases, the time gains improve as the same vegetation models can be used to classify increasingly larger sites. Algorithms which can extract traits and classify large areas are likely to improve with the increasing availability of very high-grade commercial UAV-LS surveying equipment in much the same way that SfM methods developed, beginning to rival the resolution and accuracy of ground-based TLS.


Currently, UAV remote sensing methods can only obtain above ground structural traits, and although these make up a significant component of hydrologically relevant traits, they do eliminate the collection of traits such as root structure, strength, and plant flexibility. Both UAV-LS and TLS also struggle to capture the complex structures of shrubs, with TLS requiring too many scans to resolve the structure of enough samples and UAV-LS having too low point density and canopy penetration for





such complex branching. However, methods pioneered by Manners et al. (2013) may help to overcome this by relating vertical profiles from TLS and ALS data to enable upscaling to larger extents.

### 5.3. Reach Scale Guild mapping

The benefits of remote sensing of plant traits does not come from individual plant analysis but from upscaling to larger extents. Using the same datasets provides continuity between both the individual analysis and reach wide guilds. Finding common
features of defined guilds is more computationally effective than analysing individual plants throughout the reach at present. Using structural characteristics of the point cloud alongside spectral properties across time allows the absolute and temporal patterns of each layer to enhance guild classification. It is clear that distinctive separation between guild types can initially be made on canopy height, with this providing the clearest initial separation. The need for seasonal data is emphasised by the herbaceous guilds, whereby height is a useful separator but has large variability, whereas winter and spring NDVI values are
more effective, supporting previous work emphasising the need for seasonal data to improve eco-geomorphic research (Nallaperuma and Asaeda, 2020; Bertoldi et al., 2011). Variations in NDVI were distinct between several guilds, both in absolute values and seasonal variation. Single stemmed herbs appear to be more seasonal, with lower winter values than multi stemmed herbs, whereas shrubs NDVI experience a dip in spring surveys as a consequence of flowering affecting spectral properties. When investigating differences in woody guilds, winter data collection is key, as in leaf off conditions the full tree
structure is captured in more detail and so differences in skew which are related to DBH are better captured. Later in the year, these variables become more overlapped between guilds with greater variation. Therefore, the timing of data collection will likely impact classification results, with some guilds being better separated at different times of the year. For these methods to be applied elsewhere, it therefore follows that a seasonal monitoring approach is required.

The use of random forest classification for this study site has been successful and builds on the growing body of research for their application to high resolution classifications (Adelabu and Dube, 2015; Chan and Paelinckx, 2008; Adam and Mutanga, 2009). The misclassifications from the random forest classifier are in line with misclassifications experienced by Butterfield et al. (2020) when using multispectral imagery alone, with most misclassifications happening in guilds adjacent and most similar to the true class. Woody guilds appear to be buffered by shrub guilds, potentially resulting from the image segmentation
not delineating the vegetation edge successfully. These locations are likely to have lower relative heights and so be misclassified as shrubs, whereas a better image segmentation may avoid these issues.

The resulting classification accuracy (Figure 8 and Figure 9B) shows promise for linking local scale trait modelling to larger guilds, with good separation between broad guilds and promising initial results for separation between similar guilds. The
presence of herbaceous species in the active meandering section is as expected, as these are more adaptable to changing and flood conditions whilst larger woody species are seen in more stable sections of the river when compared to the historical change, as these species require more stable hydraulic conditions (Kyle and Leishman, 2009; Stromberg and Merritt, 2016;



Aguiar et al., 2018). The classification herein advances work by (Butterfield et al., 2020) who used imagery to classify species and subsequently assign guilds, whereas this method uses the structural and spectral characteristics to designate the spatial distribution of guilds, removing the species identification component. This is important as the same species may display varying traits-based on their proximity to the channel (Hortobágyi et al., 2017) and as such, using the physical characteristics of plants can be seen as an advantage. The use of image segmentation to delineate similar areas also helps to reduce the salt and pepper effect of high-resolution data classifications and so provides an effective method when looking at high resolution structural and spectral features of a reach.

## 5.4. Eco-Geomorphic Change

Given the hydrology of the river, the majority of morphological change occurs over the winter months as expected. The temporal resolution of the surveys is not capable of picking out whether this is the result of a single flow event or continuously high flows, however it is clear that significant geomorphological re-profiling can occur within a single winter. There appears to be more localised evolution in the second winter of surveying whereas the first winter appears to show more continual response throughout the reach. The singular lower peak in water levels for the second winter as opposed to several higher peaks in the first (see Figure 1) suggests that priming may be more important for large avulsions, whereby a singular flow event of lower magnitude can incite a greater resultant planform shift. The response in summer is much smaller both in terms of deposition and erosion, with little morphological change occurring unsurprisingly. What change does occur may be from reductions in bank support from high flows leaving banks exposed to collapse (Zhao et al., 2020). The largest areas of change appear to be within the reaches absent of large vegetation, with the stable patches aligning well with those identified in the decadal analysis.

It is difficult to extract any definitive link between the types of morphological change occurring and the underlying vegetation. It is clear from the historical analysis that although vegetation plays a role in morphological evolution, it is not the sole driver of change. There are also a number of unique features in the reach which are hard to categorise or group, morphologically speaking, with different vegetation patterns, hydraulic conditions, and pre-existing morphology adding complexity. However, by grouping guilds based on their potential ability to influence vegetation, and categorising erosion and deposition into bands of morphological change in either direction, it is possible to visualise the links between vegetation and morphological change.

Figure 11 shows a bivariate classification of vegetation stability and morphological evolution. Grasses and herbaceous guilds are grouped along with bars as having the least morphological stability, followed by shrubs, and then woody guilds. Morphological evolution was split into 0 to 1 m of change, 1 to 2 m of change, and greater than 2m of change, which were chosen to represent the majority of change values. It is clear that most of the reach is shown in the lighter colour tones indicating low magnitudes of morphological change. However, areas with higher morphological change begin to become more apparent for areas of little vegetative stability, for example on the outer meander banks in several places. Darker oranges and purples





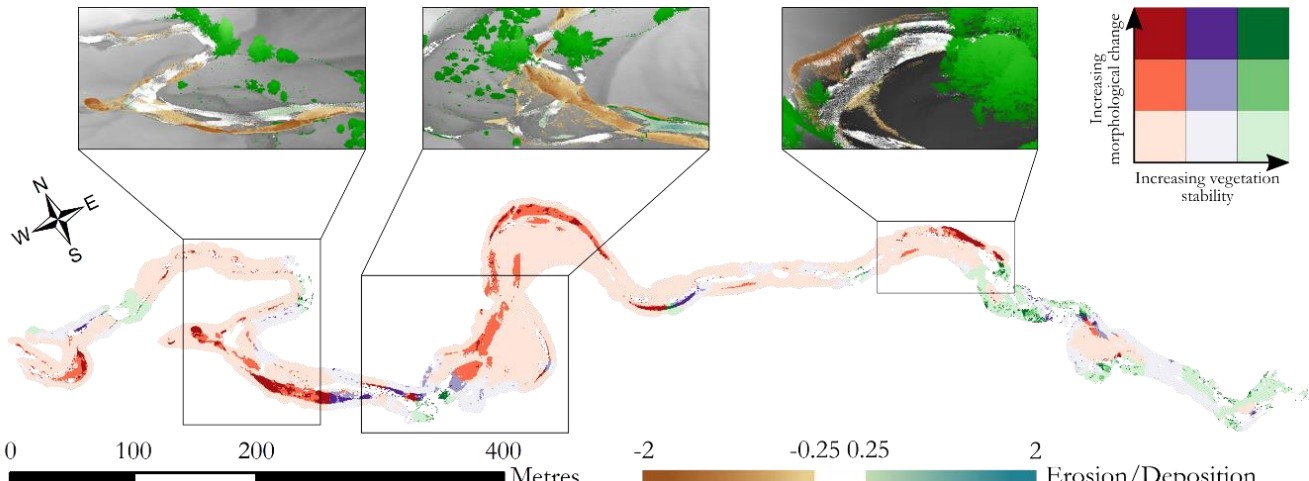

**Figure 11 Bivariate classification of eco-geomorphic process-form interactions, with examples highlighting the stabilising effect of vegetation. The bivariate colour scheme shows the impact of the likely increase in stability from vegetation (red to green) and the increasing magnitudes of geomorphic change (light to dark shades). This allows the presence and potential influence of vegetation to be mapped together. Vegetation stability was classed by grouping grasses and herbs, shrubs and bushes, and different woody tree guilds. Morphological change was split into 0-1 m, 1-2 m, and greater than 2 m of change, regardless of whether this was erosion or deposition. Insets show patterns of erosion and deposition against the presence and absence of larger vegetation across various sections of the reach.**

are dominant in comparison to the areas of dark green. Although compared to the overall areas of each vegetation stability class you would expect fewer dark green regions, there is clear evidence of light green patches where dark green patches may be expected had the vegetations stabilising effect not been present.

Some of these sections are highlighted in the panels of Figure 11 identifying regions where erosion may be expected but is not present. The left-hand panel shows a double meander bend, the first which has a heavily vegetated bank and the second which has little established vegetation. The total change in these two sections is dominated by erosion in the second meander bend which has a similar curvature to the first. The second bend does contain a knickpoint caused by overland flow which is not present in the first bend, yet the bend exit also shows far less erosion. Therefore, it is suggested that this dense patch of vegetation is having some stabilising effect, with soil cohesion increased, and flow velocities reduced. The central panel is just downstream and is constricted in planform by established vegetation, which despite substantial reworking across the survey period has remained relatively stable and exhibits deposition close to the vegetation on the left-hand bank. Subsequently, not only is the vegetation acting to stabilise banks, but likely slow the flow to encourage deposition in this area. Finally, the right-hand panel at the entrance to the region dominated by woody guilds is characterised by a large cut bank several metres in height that is progressively eroding, with the bulk of this erosion occurring just before entry into this woody guild dominated section. The vegetation on the outer bank is likely to play a stabilising role on the bank, until undercutting and removal of these trees occurs. There is evidence of such undercutting in action (Figure 12), suggesting that vegetation provides additional stability only as far as a given threshold, as was suggested in relation to the long-term decadal analysis.



## 6. Remote Sensing of Plant Functional Traits: What Next?

One of the key benefits of using remote sensing is the ability to quickly capture datasets over scales not possible with ground-based surveying. It is clear from the analysis herein that although the collection of data is fairly straightforward, the subsequent post processing time has to be taken into account. Yet once data has been processed, and the seasonality of the data acquired through spectral and structural characteristics, the success of the classification suggests that guilds can be classified for other sites that contain similar guilds, such as most temperate UK rivers which display these prominent guild types (O'Hare et al., 2016), in much the same way as other research has used previous guild classes for similar environmental conditions (e.g. Butterfield et al., 2020). It also allows guild analysis for regions that are more remote and less accessible to more traditional surveys. This improves the applicability and usability of traits methods when compared to more traditional taxonomic vegetation discretisation approaches.

Combining vegetation structural and spectral data provides the opportunity to upscale to datasets collected via other platforms, with high resolution satellite imagery and ALS datasets offering the potential to improve the impact of such classification methods. This also allows the direct measurement of trait variability rather than investigating species variability and subsequently linking these to traits. The use of purely species data may remove some of the nuance in their traits, based on location, and so limit the applicability to fluvial research. Currently, the main difficulty with traits-based analysis is getting adequate data over large enough areas, this methodology provides a potential starting point from which a set of tools to classify different hydrologically relevant guilds across larger areas can be based. This may overcome some of the scale issues in linking guilds to geomorphic change which are currently known. Currently most large-scale studies link evolution to vegetation presence, and small studies are too localised to be applicable across wider areas. This research, although not large enough to be able to link guilds statistically to morphological evolution, demonstrates that by upscaling to combine enough hydraulic and morphological conditions may allow this to be possible.

It is important to understand how the role of guilds may change. Figure 12 shows a pre and post image of the channel in this section, suggesting large scale mobilisation of large wood. Being able to identify these changes is key, because the functional role large vegetation plays in the river corridor changes depending on its life stage. Our data suggests that large trees helps to stabilise the channel, likely through increases in soil cohesion and slowing flows during the flood stage. However, once a tree is undermined by erosion and collapses to form large woody debris, it provides an increase in in channel roughness and turbulence, diverts flow, and leads to knock on morphological impacts (Jeffries et al., 2003; Sear et al., 2010). It is therefore important to consider that guilds and their influence are not stationary, but that they are dynamic through time both in terms of seasonality and life cycles. This must be considered when looking at the implications of guild dispersal and modelling, as the impact of changing from one state to another needs to be accounted for. Although the temporal evolution of guilds was not



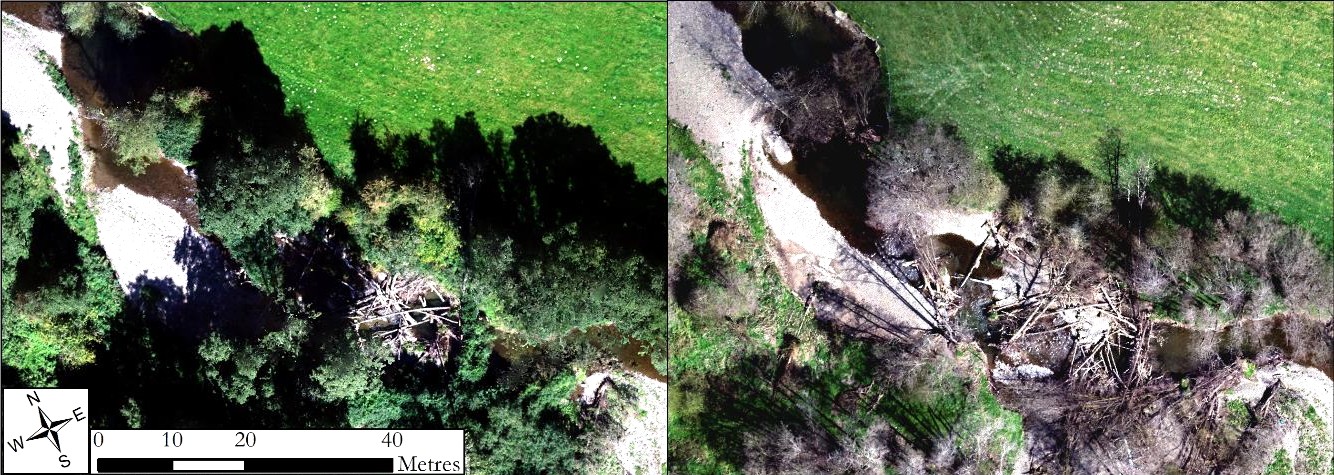

**Figure 12 The impact of undercutting within a heavily wooded reach, highlighting how vegetation and river flow interactions change through a plant's life cycle. The fallen trees create key members which form debris dams, leading to greater flow diversions, localised flooding and scour points, changing the role of vegetation from one of offering stability to inducing erosion.**

investigated, this presents itself as an area of future work, and the possibility to investigate traits-based methods to classify woody debris based on the surrounding vegetation structures.

The classification inputs predominantly focussed on structural and spectral characteristics of the vegetation; however it is widely shown that traits vary dependent on their underlying hydraulic and environmental conditions. It is therefore not inconceivable that such metrics may be used in future, such as to show inundation frequency or extent, to determine the likely composition of traits in these regions. This may take a more holistic approach and in cases where less structural data is present, allow for a more robust classification of guilds.

There are however several limitations to the methods. Variations in traits undetectable from TLS or UAV-LS methods will limit the ability to detect features for certain types of guild, such as those too small to resolve including different grasses or those with too complex structures, such as branching shrubs. Both of these are prominent features of UK river corridors and so their ommision from the analysis is a limitation. However, they can still be mapped but would require in field trait collection or species identification for the use of trait databases (e.g. TRY (Kattge et al., 2020)). The remote sensing equipment used for this research is not cheap (see Tomsett and Leyland, 2021, noting that our custom system is considerably more economical than commercial off-the-shelf packages) and requires a degree of expertise in processing and manipulating the data. However, commercial improvements are seeing more easy to deploy, cheaper, sensor systems being brought to the market, likely to have a positive impact on eco-geomorphic research in terms of allowing broad uptake of the methods developed herein for applied monitoring and river corridor management.



## 7.  Conclusion

We have presented a novel method for collecting and extracting vegetation functional trait data that is relevant to eco-geomprohic research. Herein we used  UAV-LS and UAV-MS datasets to advance our ability to collect high resolution 4D
datasets, improving the spatial and temporal resolution of riparian vegetation monitoring and geomorphic change detection, allowing us to gain an insight into how ripairan vegetation evolves and to better discretise the spatial variation of vegetation in a manner that is applicable and scaleable over large river reaches. As such, we have been able to provide insight in to how traits-based frameworks for vegetation analysis can be linked to trends and patterns in morpholoigical evolution at scales that were previously not attainable.  We have also outlined the limits for current trait extraction from remote sensing techniques.
UAV-LS can characterise larger vegetation structures and be used to upscale local TLS models, but even TLS is limited in its ability to characterise the spatial complexity of some vegetation traits at the resolution required to link with geomorphic change. This builds on current research which has analysed ecogeomorphic interactions on small river sections, or used species based imagery classification to determine large variations. The use of remote sensing allows data to be captured, analysed, related to broader dataset statistics, and upscaled to include larger reaches. Simultanously, the same data allows for the collection of
topographic responses to flow events which can be linked to the variation in vegetation. This analysis uses seasonality to improve the classification of guilds via chages in structural and spectral properties, advancing current methods available to the ecogeomrophology community. Despite some noted limitations, this research represents an important step towards better discretisation of traits across greater scales and the furthers the possibility of implementing widespread traits-based research.

Future research is needed to investigate the limits of various remote sensing methods in relation to their ability to be used for traits extarction and thereby improve understanding of a systems ecogeomorphic evolution. Of particular note is the currently untapped resource that exists in relation to coarse scale global coverage of land cover from which vegeation traits could be extracted using methods such as those presented herein to link the scales of analysis. These methods offer a bridge across sclaes, within which to consider the ways in which riparian vegetation within the river corridor is mapped, evaluated, and
modelled through time, with implications for establishing new insights into the functioning of eco-geomorphic systems across scales ranging from river sections to intercontinental basins.



**Data Availability**

The raw data used in this analysis is available at https://zenodo.org/record/5529739#.YbDLBtDP1PY

**Author Contributions**

Study conceptualisation was done by C.T. and J.L. Data collection was undertaken by C.T. and J.L. Processing and data analysis was performed by C.T., with supervision from J.L. Original draft preparation by C.T. with reviewing and editing by C.T. and J.L. All authors have read and agreed to the published version of the manuscript.

**Competing Interests**

The authors declare that they have no conflict of interest.

**Acknowledgements**

This research was funded by the Natural Environment Research Council (NERC), grant number 1937474 via PhD studentship support to CT as part of the Next Generation Unmanned System Science (NEXUSS) Centre for Doctoral Training, hosted at University of Southampton.

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
