# Peer review of "Exploring the 4D scales of eco-geomorphic interactions along a river corridor using repeat UAV Laser Scanning (UAV-LS), multispectral imagery, and a functional traits framework."

_Earth Surface Dynamics, 2021_

## Author Comment (AC1)

Dr Christopher Tomsett School of Geography and Environmental Science University of Southampton Southampton SO17 1BJ

14th September 2022

ESURF-2021-102: Exploring the 4D scales of eco-geomorphic interactions along a river corridor using repeat UAV Laser Scanning (UAV-LS), multispectral imagery, and a functional traits framework.

**Response to Comments from Reviewers**

Dear Lina Polvi Sjöberg and the Two Anonymous Reviewers,

We thank the Earth Surface Dynamics editors and both reviewers for their constructive and helpful comments on our submitted manuscript. In addition, we thank the editors for allowing us additional time to complete the requested revisions. We have made extensive changes to the manuscript, including some additional analysis. Below we outline how we have addressed the reviewers overall feedback and the individual comments for each of the reviewers in turn. For clarity, we summarise the review comments in **bold and italics** and list our response below each one, highlighting the line numbers in the new manuscript where changes have been made:

**Reviewer 1:**

I believe that the manuscript by Tomsett and Leyland will, with some refinement, make a significant contribution to the ecogeomorphic literature. While the paper is technically robust, using a variety of tools and analyses to link remotely sensed data to hydraulicallymeaningful vegetation characteristics, there are a number of deficiencies in its current form. For one, the need for and novelty of this work needs to be more carefully constructed. To develop a spatially robust understanding of plant-flow-sediment interactions, this work comes at the problem by way of characterizing plant morphology and classifying groupings of plants based on like forms. This is in contrast to Diehl et al 2017 and Butterfield et al 2020 who instead use the identification of species, and the link between species and their functional traits, to classify groupings of plants. These are fundamentally different ways to approach this problem, creating a different product. My sense is that the authors are very aware of these differences, and reference it throughout

**the paper, but the differences, and strengths of the different approaches should be highlighted and made clear in the introduction.**

We have completely reworked the front end of the paper to better point out the gaps in the literature and to highlight the potential of our work in comparison to the work already undertaken in this field. This has subsequently been intertwined further throughout the manuscript to highlight the benefits and drawbacks of each approach to this problem. The introduction concludes with an adjusted tightly defined set of aims which seek to highlight the novelties of the work and point to some new analysis of seasonal vegetative excess drag that we have introduced in response to comments below. It is hoped that this makes the distinction between our work and the current research so as to make it clear to the reader where the differences lie.

Because the approach presented in this paper does not use any ecological observations nor actually link their prediction of functional types to topographic change, the approach in its current format does not make the linkage between ecological and geomorphic processes. The authors acknowledge this in the discussion (lines 820-825), but again, this should be brought up to the introduction. The approach described here seems to have two major benefits over Butterfield et al (2020)'s use of classified remote sensing imagery as a way to create maps at scale: 1) there is not a need to have field-measured traits to identify a species' functional type and 2) this approach can add a fourth dimension, time. The authors discuss #1 as a rationale, but #2 is poorly developed and executed. They discuss the importance of time-or seasonally- varying parameters for understanding plant-flow interactions and use seasonal differences in NDVI as a way of differentiating between different types of herbaceous plants, but do not provide any meaningful way of characterizing or classifying the differential impact of plants on fluvial processes during different seasons. For example, is the difference in seasonal NDVI between branching and single-stemmed herbaceous plants hydraulically meaningful? Could you develop one map of functional types for the winter and one for the summer? Also, its not clear as to if the finding of a different spectral signature between the two herbaceous guilds will hold for other settings, or is it because of the difference in species type here?

We have sought to address this weakness by extending the analysis in two ways. Firstly, functional group maps have been created for each year of the analysis, rather than producing one. As the seasonal variation is required to achieve this mapping technique, only one map per year is used to show the distribution of these groups. Secondly, to account for seasonal variation that is not captured by these spatial distributions, calculations of associated excess drag for these functional groups at differing stage levels has been undertaken, to better understand the likely changes in seasonal excess drag across the domain. We used Delft3D to recreate a reference flood through the reach and used the resultant flow metrics, together with measured frontal areas across time, to calculate excess drag for each functional group.

We have added extensively to the analysis to calculate the impact that seasonality has on measured vegetation structure, thereby showing the ability of our methods to discretise subtle changes through a combination of 3D structure and spectral response. We feel now that we much better capture the potential of our developed methods to assess temporal and spatial

variability, showing how in this case different functional groups can be linked with areas of greater morphological change.

As we recognise the message from both reviewers to better incorporate and expand upon the temporal component of the work, the introductory sections, methods (e.g. complete new section 3.4), results and discussion have all been reworked to better incorporate the temporal component of change and the seasonality component better.

In addition, we have made Figure 6 (the reach classification) a two panel figure to show annual change in functional groups and attributed morphological change across functional groups through the reach to seasons in a new Figure 9, which offers detailed insight into the links between functional groups and morphological change. This in effect deconstructs the histograms of change in Figure 8 in order to see in which functional groups this net change is occurring most. Another new Figure (10) shows the dominant changes in functional groups/land cover across the survey period, so that the changes seen in the two panels of Figure 6 can be better highlighted, and used to explain changes in reach excess drag in Table 3.

Finally, Table 3 and new Figure 11 attempts to show the detail that our approach can offer by estimating the excess drag of different functional groups exerted through time (Table 3) and on different hypothetical flow depths across the flood plain, simulating the real world impact of vegetation across the riparian corridor on in-channel and overbank flood waves. This is hoped to help both address the temporal component of each vegetation functional group across the survey period, and also to highlight how the spatial analysis allows depth dependency to be introduced to help untangle the aggregated nature of eco-geomorphic feedbacks.

**As the authors think about how to more carefully frame their work, one additional consideration is the more precise use of terminology. The idea to adopt concepts from ecology into geomorphology as a way of investigating the interactions is welcome and represents a promising path forward in integrating ecological and geomorphic processes. However, I found that the authors use terms such as "traits", "functional types", and "guilds" fairly loosely. Some specific examples are provided below.**

This has been addressed by rewriting much of the background to remove extra information and by creating a clearer narrative with the consistent use of terminology, especially in those areas highlighted below (see separate responses). We have now purged the use of guilds and use only functional groups, with a brief explanation for this in the introduction section, specifically relating to the different uses of terminology by Blondel (2003).

The paper uses a variety of datasets and analyzes them in technically sound ways, but there are numerous missed opportunities to take the data one or two steps further to provide a little more insight into the ecogeomorphic value of the classification system. The stated goal of linking traits-based guilds with ecogeomorphic change and capturing the temporal variability is not quite accomplished with the current analyses. My best understanding is that the authors use the long term analysis to link veg/no veg with bank erosion and the likelihood of avulsions. While this is an interesting analysis, it does not provide any details on the importance of functional groupings of plants on morphodynamic processes, nor provide insight into the change in plant-fluvial process interactions with season. Instead, can you create functional plant grouping maps for each of the four topographic change maps (Figure 10) and evaluate the relationship between erosion, deposition, or no change and functional group? Even if you cannot create unique classification of plants for each change map, assuming the distribution of plants remains the same (OR creating two classification maps- one summer one winter), can still give you some powerful data that can help achieve your stated goals to your "Aims" in section 1.5.

As noted above, we have completed extensive additional analysis and created multiple new figures and discussion which seeks to address this point, with specific reference to the revised Figure 6, new Figures 9 and 11, and Table 3.

Both of these address the gap between linking annual distributions of the functional traits and there aggregated changes in morphology (Figures 6 and 9) as well as using the seasonal components of excess drag to identify patterns in erosion during a specific winter period. This specifically answers our aims 3 and 4.

**There is little validation in this paper to help the reader understand if this approach is helping to advance ecogeomorphic studies in a meaningful way. You must have a sense of the types of species growing at the site. If so, you should provide the reader with a summary of these types of communities and consider comparing the measured traits with traits listed in the literature or in the TRY database.**

We thank the reviewer for this suggestion, providing an extra source of validation for our methods. Section 4.2.1 has been extended to include explicit comparison with data from the TRY database, as well as from the wider literature, showing a good level of agreement with our extracted metrics where they could be found and compared. It also helps to highlight the issue with current databases, whereby some species have limited data recordings of the traits highly relevant to hydraulic interaction, or in some cases none at all.

**Specific Comments:**

**Line 1: In its current form, the title leads the reader to believe that the analyses in the paper evaluate the temporal dynamics of ecogeomorphic interactions.**

We believe that in addressing the reviewers comments we truly now address the stated aims to explore 4D eco-geomorphic interactions.

**Lines 16-19: If I understand this correctly, you used the long term analysis of channel changes and a general classification of "trees" vs "no trees" to come to these conclusions. If so, it would be more accurate to say "We show that vegetation generally has a role in influencing morphological change through stabilization...."**

This section of analysis has subsequently been removed from the paper in line with comments from Reviewer 2, and as such is not included in the abstract.

**Lines 44-46: Traits-based classifications are intended to achieve this, if one can link fieldmeasured traits to species/functional groups.**

**Lines 45-49:**

Sentence structure changed in order to shift emphasis onto limitations of data collection as opposed to the traits based methods.

**Line 55: "how vegetation is modelled" is vague. Instead specify the ways in which people model vegetation? Bulk roughness? Cylinders? Rigid vs stiff?**

**Lines 59-60:**

This sentence has been restructured to specifically mention the modelling methods tested in the referenced literature, referring to how vegetation is representation rather than modelled.

**Line 59-61: Are you referencing aquatic vs riparian (or terrestrial) vegetation here?**

Line 65:

We have adjusted this to state that it refers to terrestrial vegetation that in the event of a flood would be submerged by flow.

Sections 1.2 and 1.3: These sections need some work to provide proper background on plant traits, their use in ecogeomorphic studies, and how existing approaches are not adequate. I found the explanation of hydraulically-relevant traits to be scattered and if I was not familiar with the literature would be lost as to what a hydraulically-relevant trait is and why its relevant. You might consider referencing Table 2 in Diehl et al 2017 and briefly describing the different traits. This will then help set the stage for Section 1.4, which should focus on how to measure these traits using remote sensing- the challenges and opportunities.

We agree that these sections were a little muddled and we have undertaken an extensive rewrite to provide a clearer narrative surrounding hydraulically relevant traits and why they are different to functional traits as described in ecological research, citing the Diehl paper as suggested. Section 1.4 has also been edited in order to focus on the ability to obtain traits through remote sensing, as opposed to focussing on the remote sensing of vegetation in general.

**Line 70-75: I may read this incorrectly, but generally functional traits are used to define a functional group and so the argument is strange to me. This is different from either a species-specific or typological approach because functional types are groupings of species (likely typologically similar) with similar responses to the environment and with similar effects on ecosystem processes.**

In the reworked paper this paragraph is no longer deemed to be needed and so has been removed to avoid confusion. As above we agree that some sections were muddled and may have led to some confusion.

Line 81-82: Here it seems like you jump from traits to functional groups. The benefit of a functional group approach is the ability to generalize. If you were to take a traits-based approach alone, you would create maps of essentially different physical characteristics-say one of height, one of frontal area, etc. This would be informative, but not helpful in understanding the plant's full impact on the environment. Functional types clusters or groups plants with similar arrays of traits that, in the aggregate, explain the response (or impact) of that plant type to (on) its environment.

We have rewritten much of this section, to improve the overall narrative around the transition from traits to functional groups, and why using this grouped approach is both helpful in understanding what is happening within an environment and how it can be applied elsewhere. As a result, we believe the transition from traits to functional groups is better explained and outlines our reasoning.

**Line 97: Given your description in the following sentences, it might be more appropriate to change out "hydrological conditions" to "environmental conditions".**

Line 111:

Terminology changed as suggested.

Lines 105-109: This point, that there is a lot of variability between species needs to be more carefully flushed out if it is one of your major points and rationale for your approach (vs starting with maps of species tied to traits). The traditional ecological traits-based approach relies on the fact that the traits used to define functional groups should have greater variability between species than within species. This comes up again in lines 132-134.

As with the majority of this section, this has been reworded to highlight that differing traits do occur within the same species depending on their environmental stressors, and that this highlights the need for a mix of taxonomic and traits based methods.

**Line 116-117: This seems out of place.**

Removed to improve the flow of this section as suggested.

**Line 135: Do you mean "Hydraulically Relevant Functional Traits"?**

Amended accordingly.

**Lines 162-166: These two sentences seem to contradict one another.**

Lines 160-165:

This section is included in the parts which have been largely rewritten to provide a clearer introduction. The intended aim of this sentence was that although density has been shown to be important, the distribution of the plants may be more so. As remote sensing can help to achieve this, it was highlighted here and in the reworded section.

Lines 184-186: The height of a plant during submergence is not a trait. Instead, it's a function of the plant's height and flexibility, and maybe also other factors that determine a plant's pronation (e.g., branching structure, leaf area). This is an example of where you need to be careful with terminology. Also, the introduction of temporal variability is potentially critical to your framing of a need for 4 dimensions, but buried as an aside in this paragraph.

It was not intended to suggest that the plant height at time of submergence was in itself a trait, but more to highlight that using traits databases can have limitations. To address this the sentence has been rephrased to suggest the importance of using a combination approach of databases and remote sensing/field investigations.

**Lines 210-215:**

The final sentence has also been removed and placed at the end of this section in a more explanatory paragraph to flesh this out a little, with the aim of specifically highlighting the relevance of 4D, as was also noted by Reviewer 2.

**Line 226: Who operates this gaging station? Where did you download the data from?**

Information regarding the source of this data has been added to the figure 1 caption, stating that it is from the UK Environment Agency.

**Lines 287-308: Cut this section down, relying on the fact that there is a published paper. For example, there is not necessarily a need to tell the reader of this paper about the battery life of the UAV's.**

This has now been considerably shortened in line with the suggestions, only leaving the sentence pointing to the supporting paper, and with suggestions from reviewer 2 has been compacted to include extra details on reference systems.

**Section 3.3: This is a cool methodology**

We thank the reviewer for their supportive view on our new method.

**Lines 342-344: Not sure what "a traits-based rather than bulk roughness approach is likely to be limited."**

**Lines 298-302:**

The wording of this has been revised to explain that both grasses and shrubs are difficult to extract traits from using remote sensing methods, and as such traits based approaches have their limitations here, so the processing is limited to excluding these groups on the whole.

This is however not the case for identifying excess drag, whereby frontal areas of the vegetation can be detected, and so is included in the analysis.

**3.3.2: Would be helpful to list all the traits you extract, or create a table. Why didn't you measure plant density? That is one that could be accomplished through remote sensing, can be important, and will vary in different parts of the river.**

**Line 305:**

The traits collected are now explicitly listed at the top of the section, to make clear what the purpose of the following method is. This will clearly outline to the reader what parameters are being measured.

Density was originally measured but not included in the analysis as this would not be an average density but a singular value which would not be useful for the PCA analysis. However, with the inclusion of the spatially varying excess drag calculations it has now been incorporated in line with other review suggestions and a description included herein in new section 3.4 (*Lines 470-476*).

**Line 384: Might be helpful to create a table of the guilds you adopted from Diehl et al 2017.**

Line 348:

This has now been included as a list of functional groups being used at the beginning of the functional group identification section. The aim of this is to both inform the reader of what these groups are explicitly, before then explaining how these were chosen and why. This seemed a more fluid way of incorporating the groups as opposed to a table, as with the above comment. They are also subsequently listed in table 2 when discussing reach scale analysis, and so avoids duplication.

**Line 385-386: What are "bulk roughness metrics" and how were they applied?**

This is now explained in a subsequent section and so has been removed to avoid confusion. This was originally intended to denote that assumptions were made based on the group as a whole, rather then the accumulation of individual plants.

**Line 390-392: How did you handle woody seedlings and saplings that might be a similar height to herbaceous plants?**

Unfortunately, due to the resolution and nature of the remote sensing data, this separation could not be made, to be more clear about this, a discussion point has been added in section 5.1. on trait extraction to comment on the challenges here, the potential drawbacks, and possible solutions/research avenues (*Lines 788-794*).

**Line 514: Change modelling to modelled**

Changed.

**Line 556: This is the first time you bring in elevation as a "trait" to classify guilds. Is this value measuring the elevation of the ground surface around the plant?**

**Line 566:**

This has been corrected and was an oversight in terminology, elevations in this context does indeed refer to the canopy height above ground level as in Figure 5. Two more uses of the word elevation have been adjusted to 'height' to make it clearer that it is relative to the ground surface and not above sea level.

**Lines 725-729: I get that this is one of the main benefits of this work, but by taking out species consideration, you remove the capacity to evaluate the full set of feedbacks among environment-plants-topographic change and in essence you are just creating a map of plant characteristics.**

In order to emphasise the use of species identification still in this research area, we have included extra clarification throughout to highlight the need for species information where possible, whilst also pointing out some of the drawbacks of a species identification only approach.

It could be argued that our approach is in essence various layers of plant characteristics, yet by grouping them into areas of similar characteristics this is part of the functional groups process. It is hoped that we have provided balance between the two approaches, and where appropriate highlighted the benefit of taxonomic methods over remote sensing methods for assessing ecogeographic interactions.

**Figure 11: Was this figure, and the matrix, created by comparing your guilds with topographic change? Or was it done conceptually? Again, I am not sure why you didn't perform a more comprehensive analysis of the differences in topographic change in and around the different guilds over different seasons.**

This figure has been removed and replaced in the revised manuscript to shift the emphasis onto the temporal component of change as requested by the reviewer and outlined in our responses above. This is in the form of Figure 11, which both relates excess drag at different depths to morphological evolution for each of the functional groups. Along with the seasonal analysis, we believe that we have improved the manuscript to account for the temporal and spatial complexity of vegetation monitoring.

**Reviewer 2:**

This paper presents an intriguing and likely novel data set, with multiple repeated highresolution scans of a vegetated floodplain using numerous different cutting-edge techniques to assess the vegetation structure and therefore roughness. Vegetation classifications near a highly mobile river reach are performed using machine-learning techniques that leverage modern algorithms and computing power. However, despite the numerous data sets presented here, the manuscript does not yet sufficiently justify how it represents a substantial contribution to scientific progress. Most obviously, the paper claims to be a 4D (3D space & time) comparison, but it falls short of this intent. For one thing, remote sensing data are processed to provide static 2D (rather than time-varying 3D) maps of vegetation guild coverage (e.g., Figure 4A). For another, although geomorphic channel change is characterized, it does not appear that temporal changes in vegetation are considered. There are undoubtedly changes in vegetation phenology (flowers vs. no flowers, leaves vs. no leaves) and morphology (herbaceous shoots vs. dry stalks) over time in vegetated regions, as well as growth of new plants and shoots, but this is not featured; instead, discussions of change over time focus on unvegetated fluvial regions. In fact, it is unclear to what extent 2D maps, let alone the location and characteristics of individual plants, are consistent from one time to another. Given the focus of the paper on changes in vegetated regions, it is a major oversight to omit a detailed discussion of differences (due to changes or uncertainty or both) between repeat scans in regions that remained vegetated (no avulsions etc.).

In line with the above comments, we have undertaken extensive revisions to the paper to justify how this work has provided scientific progress. Initially, we have made adjustments to the way in which data classification is undertaken, to use both leaf on and leaf off conditions for each year to create maps of functional groups (formerly referred to as guilds) for both 2020 and 2021. Although this is represented in a 2D classification, the data behind this classification method are inherently 3D dimensional as they are representative of structural properties that are devised in 3.2.2, and as this is performed through time, is subsequently referred to as 4D. It is hoped that the revised wording throughout the methods and results helps to establish how this is the case.

The resultant morphological maps are now relatable to changes in spatial extent of these vegetation classes with the inclusion of a double panel classification figure (Figure 6). Although there is variation in leaf on and leaf off conditions in the time period, these are not shown through this classification figure, but through the undertaking of new analysis such as identifying spatially varying excess drag for the reach, comparing this to high flow events, and the resultant erosion and deposition.

A discussion of how the coverage and spatial extent of these functional groups has also been included, to indicate dominate changes as well as how this effects excess drag, highlighting spatially relevant change.

Second, the manuscript examines two drastically different spatial/temporal scales of interest, with only loose connections between them. One scale is the decadal scale of channel change and avulsion (Sections 3.1, 4.1); the other is the seasonal/annual scale of individual plant growth and characterization (Sections 3.2, 4.2). Given the extensive discussion of the hydrodynamic impacts of vegetation that was presented in the introduction, as well as the highly resolved tree-level observations possible with the remote sensing detail, the manual classification of the floodplain into only two vegetation classes (large vegetation vs. not large vegetation) for erosion assessment is massively simplistic. The spatially explicit location of erosion and new channels during the study year are presented (Sections 3.3, 4.3), but these locations are compared only anecdotally in Figure 11 to the various types of vegetation that were identified throughout the study reach. Without some attempt to quantitatively tie these various types of data together, the paper lacks cohesion, as well as misses its opportunity to evaluate the geomorphic importance of its classification scheme as well as controls on channel change.

We have decided to remove the decadal analysis of change altogether based on the review comments and the already lengthy nature of the paper. On balance, whilst the aim was to provide some useful geomorphic context of the reach, we decided that it was better to address the concerns surrounding the 4D nature of the work. As a result, a number of the following comments relating to this section of the work have not been responded to.

Third, although the focus of much of the article is on traits-based classification of vegetation, no validation data are presented for this site or even these species. Without some sort of independent assessment (ideally from field observations), it is difficult to know to what extent the categorization presented herein is appropriate. Previous studies (e.g., Butterfield et al. 2020) have included ground truthing. The algorithms that were used were developed for different species in different ecological settings (e.g., Scots pines in Finland, beech and oak in the Netherlands), so it is difficult to assess site-specific validation, especially for application to non-woody grasses and herbaceous plants. An error/misclassification analysis based on field data (which may have been obtained – cf. Line 407) would greatly enhance the vegetation classification portion of the study.

Some of the issues raised in the above comment have been responded to based on comments from Reviewer 1. Overall, we have addressed the above issues by using field photos to identify species that are present at the site, and which functional group they belong to. These species were then used to undertake a search of the TRY database of traits and wider literature for reported values that were extracted in this study. Although limited by data availability, these values corresponded well to those collected in the field, and with the accuracy assessment provided in Tomsett and Leyland (2021) suggest that the data is accurately reflecting the real world vegetation. The algorithms were developed based on a sample of species as mentioned, however the use of QSM extends beyond these species and provides a good theoretical basis that they should be applicable to vegetation across different scales assuming appropriate parameter values are used. As a result, with the comparison to observed values in the literature we are confident that the methods are appropriately used.

A misclassification analysis is presented based on high resolution imagery and site knowledge within 4.2.3 and presented in Figure 7B, and despite not being based on locations based in the

field, uses a combination of this high resolution imagery, field notes, and photos to support the determined 'true' classes.

**Specific Comments**

Line 42ff: The introduction focuses on the classification of vegetation into a relatively new framework developed to characterize eco-geomorphic relationships. Though this is an intriguing question, this narrow focus likely represents a missed opportunity to provide enough detail that ecologists and biologists could appreciate and use the results. An expansion of the idea of "traits-based classifications" to include other ecological goals may make this paper much more useful to a broader group of readers.

**Line 47:**

A small amendment to paragraph and sentence structure now incorporates a reference to the benefits of any methodological advances helping those outside the fluvial domain also. We accept that in order to be wide reaching and benefit those outside the community it is important to provide enough detail for the study to be useful. However, care was taken not include too much wider context outside of the fluvial eco-geomorphic research due to the already lengthy nature of the paper, which does have a highly fluvial focus.

**Line 55ff: The section on "The importance of vegetation" is focused exclusively on the role of aboveground vegetation in affecting river corridors. Surely the roots (belowground portions) are important as well. Although these portions obviously cannot easily be measured by remote sensing, their known contributions should at least be summarized.**

**Lines 76-83:**

It is agreed that the below ground vegetation is of great importance and should be summarised briefly within this section. This aspect of the section is expanded to highlight this and also comment briefly on the difficulties of obtaining such data.

**Line 164ff: An important aspect of vegetation reconfiguration and drag is whether the stem is woody or not. A discussion of this aspect (and relevant citations) should be added to this section on functional traits.**

**Lines 149-153:**

This has been included in the discussion around flexibility, and that taxonomic approaches may be suited to determine the extent to which a plant is 'woody'. It is also included how future studies may be able to use remote sensing imagery to identify species to infer woodiness.

Line 170ff: To completement the extensive discussion of the impact of vegetation on hydrodynamics, the background information on the impact of vegetation on scour should be increased, especially at the scale of the bar or channel, which is what is measured in this study. This crucial paragraph does not contain any in-text citations, despite a wealth of experimental and field studies on the topic. This paragraph should be expanded and should include specific citations to previous studies.

**Lines 165-170:**

This section has been added, within the context of aggregated vegetation dynamics and how this impacts morphology, before commenting on the link to traits based methods. This has not been extensively reworked to limit the word count in the introduction and background, but has included a number of key references in to field and flume studies on vegetation and morphology dynamics.

Line 176ff: The subsection titled "Remote Sensing of River Corridor Vegetation" is quite short and does not do justice to previous attempts to remotely sense vegetation that may be present in river corridors. A key omission is a description of previous efforts to use UAVs and TLS for remote sensing of vegetated regions, especially their methods (i.e., indices/proxies used, ground-control points, SfM, etc.) and successes and failures. Here are a few papers that might be relevant:

- Calders, K., Adams, J., Armston, J., Bartholomeus, H., Bauwens, S., Bentley, L. P., ... & Verbeeck, H. (2020). Terrestrial laser scanning in forest ecology: Expanding the horizon. Remote Sensing of Environment, 251, 112102.
- Martin, F. M., Müllerová, J., Borgniet, L., Dommanget, F., Breton, V., & Evette, A. (2018). Using single-and multi-date UAV and satellite imagery to accurately monitor invasive knotweed species. Remote Sensing, 10(10), 1662.
- Müllerová, J., BrÅ-na, J., Bartaloš, T., DvoÅ™ák, P., Vítková, M., & Pyšek, P. (2017). Timing is important: Unmanned aircraft vs. satellite imagery in plant invasion monitoring. Frontiers in Plant Science, 8, 887.

This section has been changed to trait data collection in order to better summarise the contents of the section which is trying to evaluate trait data collection and extraction from ground and remote sensing methods, as opposed to broadly looking at vegetation and remote sensing. As there are several papers in multiple disciplines covering the use of UAVs and TLS for environmental data collection covering SfM, GCPs etc, we do not feel it necessary to cover this here and further increase the length of the paper. Amendments have been made to better construct this section, highlighting the fact that trait data collection remotely is still in its infancy, yet there is the possibility for it to be undertaken. We also include some additional references on traits from outside the fluvial domain that have utilised remote sensing, also helping to broaden the appeal of users as raised in the first specific comment.

**Response Grouped for the following comments:**

- Line 236: Specify in text how bank edges were digitized and, if manually, then at what precision.
- Line 246: Explain in text exactly how a mix of spectral bands were used to highlight channel position of banks under trees, or provide a citation for this method.
- Line 254: Specify in text whether the same centerline/transects were used for each digitized year or whether these changed position each time, and, if the latter, how this horizontal change affected assessments of channel width.
- Line 255: Specify whether the SCE was calculated separately for the left and right bank.
- Line 260: Specify in text whether the woodland areas needed to be near the channel. Also clarify whether there were changes in the distribution of large vegetation over time and, if so, how that affected the classification: i.e., if vegetation grew in a region over time, was it classified as vegetated, or not, or did its classification change over time? Somewhere (Figure 1? Figure 4? Figure 8?) a map of these classifications should be shown.
- Line 263ff: "the analysis was repeated for changes..." The rest of this paragraph is unclear. Be specific about what happened: what does removing a transect mean, or using a separate baseline? What does baseline mean in this context? Without making this point clear, the assessment that "the impact on the results from channel switching can be isolated and removed" is not supported.
- Line 268: Specify what statistical comparison was used. Either a t-test or a nonparametric method should be used to evaluate differences between groups.

The section to which these comments refer has been removed in the revised manuscript.

Line 310ff: Specify how/whether analysis was performed for each of the flights shown in Table 1. Were data sets projected to a common reference frame/grid, or did they differ? Were each of the five identified steps performed independently for each data type (UAV-LS vs. UAV-MS vs. TLS), or did some steps involve the comparison of multiple data types? Were repeat scans of the same area processed completely independently, or (for example) was the spatial location of a vegetation point cloud (i.e., specific plant) identified at one time used to identify a point cloud location at a different time? Did all analyses require classification into individual plants, or were some vegetation types best classified using bulk metrics (canopy height, density, etc.)? Answers to these basic questions are important for interpretation of the rest of section 3.3.

A common reference frame that was used throughout has been included in the data introduction. The subsequent methods have been revised, however we have respectfully declined to include all of the requested detail here as we feel that the methods are quite complex and do not want to confuse the reader. However, as there are numerous datasets across a lengthy timescale which may be tricky to keep track of, we have made more effort to explicitly mention exactly which dataset types are being used at different processing stages, and the temporal context in which those sit (*e.g. Line 278*).

**Line 323: "leaves and flowering parts were removed from the clouds..." How were these items identified, and was it performed only for TLS or all studies?**

**Line 278-280:**

Comments as to how these were identified and the data source is re-emphasised from the preceding paragraph to show what methods were done to achieve this and that this was performed on the TLS data.

**Line 325: "Any statistical outliers were detected, removing points 2.5 standard deviations and above the mean distance between points...." How were distances between points calculated, and what does it mean to remove a point that is above a mean distance?**

**Line 282-283:**

Distances were based on the average distance between points in the segmented cloud, and then the distribution of these mean distances are used to remove points that are greater than 2.5 standard deviations above the mean. Removing a point simply states that it is no longer included in the analysis. This has been incorporated in to the methods to be clearer about how this process works.

**Line 326: "...a dataset consisting of 37 herbaceous plants." There were presumably many more than 37 herbaceous plants within the study site. How were these 37 selected? Was it the same plants for all repeat studies, or did they change over time?**

**Line 284-285:**

Only one survey was undertaken with the TLS, which is now clearer with some of the changes to the methods overview, so all samples were from the same time period. It is now mentioned how those selected represent a sample and were chosen based on complete vertical profiles and from across the study area.

**Lines 341-342: "Shrubs and grasses who structure could not be fully resolved from the UAV-LS or TLS data were not analyzed for traits extraction." This seems like a major hole in the current analysis, which set out to characterize all types of vegetation.**

Whilst it is not possible to use UAV-LS or TLS in this analysis to extract short grasses structure or shrubs and bushes such as hawthorn, we do continue to include them in the analysis and classification of vegetation, being defined as their own functional groups. This is added to by the calculation of excess drag using frontal area, for which shrubs are included. It is also not an explicit aim to characterise all vegetation in the reach, but those whose traits can be extracted from remote sensing data, and we acknowledge in the work the limitations of using a purely remote sensing based approach.

**Line 394ff: It is unclear for which/how many plants/scans the PCA analyses were performed, and whether these were the same among different methods (TLS vs. UAV-LS vs. UAV-MS). Clarify in text.**

Additions and more specific references to source datasets within 3.3.3. have helped to identify how the PCA was performed, and show that two separate PCA were performed across herbaceous and woody groups, and the reasons as to why this approach was taken.

**Line 407: "field observations" – explain how and when these were performed.**

**Lines 371-372:**

These were simply notes of the field site and photographs taken on site which were to be used for species identification. This has now been stated in the text, along with when these notes and photographs were taken.

**Line 445: "Due to the limited number of samples being used, ..." An error analysis is important. If not enough samples were used to enable even an internal consistency check, then the number of samples should be increased.**

This is a drawback of the current dataset. Processing individual vegetation samples and creating their individual traits is currently a time intensive process, and a limit of the number of samples to be used had to be made. In turn, this affects the number of training objects created in the image segmentation process.

The classifier itself can give an idea of accuracy using the OOB methods described in this section, whereby the forest is constructed by using a subset of the samples to then test against the excluded samples, and as such performs and internal accuracy assessment.

As well as this, we utilise a follow up accuracy assessment using a number of points throughout the study area that have been classified based on high resolution imagery as referenced in response to reviewer 1 and also in the third main comment of the review. We deem this to be a sufficient attempt to quantify the accuracy of the classification given the limitations of the dataset.

**Line 464ff: Explain how SfM and UAV-LS data sets were combined. Was a single DEM produced for each observation date? Etc.**

**Line 438:**

A clarification on these being done for all dates is included, with a reference to how this was done already included.

Line 482 Table 3: Provide statistical significance for differences between each classification. Remove bonus "s" from caption. Specify units for data shown in table.

**Line 495 Figure 4: In Panels B and C, bars should show some sort of uncertainty, stemming perhaps from the horizontal accuracy of transects or bank determination.**

As in above sections, both of the above comments refer to sections that have subsequently been removed from the manuscript.

**Lines 518-519: "Overall, model repeats appear to have good agreement with one another, and provide a basis for separating out vegetation with similar hydraulic functional traits." Do these model repeats refer to repeated classification of the same image, or comparison between images? Add this information to methods, and also explain here.**

This sentence refers to repeats of the same plant being modelled 10 times as described in the methods section (3.3.3), this having being amended to highlight the individual plant specimen being repeated this 10 times. To continue this reaffirmation, reference to individual plants has been included in the rewording of this section (*Line 489 onwards*) as well as reference to repeats of individual models to make this as clear as possible to the reader.

**Line 521 Table 4: If six vegetation classes were used, then this table should show all six vegetation classes, as well as a statistical analysis of whether values are the same among different classes. In caption, specify meaning of all initialisms.**

Table 4 has since been removed to improve the narrative and flow of this section. This now better details the consistency in the modelling through the QSM procedure, and gives more detail as to what the relative size of the variation for different values. It is hoped that this is more informative for the reader, and avoids any confusion in the number of classes used. As only four of the six functional groups underwent this analysis as outlined in the methods, the QSM modelling can only have results for these four classes.

**Line 556: In text, explain whether any attempts were made to characterize the understory vegetation. Unclear as written.**

The decision not to characterise this was due to point densities below the canopy, however, references to this limitation have been added in to the discussion to comment on this and the potential limitations (*Lines 823, 883*).

**Line 569: "....many areas being classified as expected." On what basis were these expectations made or assessed?**

This wording has been removed. It originally referred to the resulting accuracy assessment performed, but with an indication of the qualitative assessment of the distribution in the

paragraph. This has also been amended with the inclusion of annual vegetation maps, which incorporate the temporal aspect as outlined at the beginning of the review.

**Line 590 Figure 8: For which time period was this classification produced? If only produced once for the entire period of study, then how much change was observed during the study?**

In line with the overall methodological adjustments made to the paper, there are now two classification maps shown, one for the first year winter/summer cycle and one for the second year (now Figure 6).

**Line 633ff: "It is not possible to isolate a single variable that may cause such switches to take place, such as particular flow thresholds, baseline conditions, vegetation, or soil characteristics." It does not appear that any detailed, let alone quantitative, analysis of any of these factors was performed; without that analysis, it does not make sense to comment that no such factor was identified.**

As above, this section of methods and analysis has been removed from the manuscript.

**Line 654: "...especially once trade-offs in terms of time and spatial extent are accounted for." This is an intriguing idea; would be nice to see it expanded.**

This has been developed to give an idea of how field surveying when applied to numerous areas may be less effective than remote sensing based data extraction, especially in regard to being able to make these methods more widely used both within the fluvial and wider ecological domain.

Lines 739-741: "The largest areas of change appear to be within the reaches absent of large vegetation, with the stable patches aligning well with those identified in the decadal analysis." First, as noted above, the polygons showing the spatial location of large vegetation are not shown anywhere in the manuscript. Second, comparing Figures 4A and 8, it appears that the downstream mobile bend was located within a reach with large vegetation. Be more specific (and ideally more quantitative) with how documenting how the results were used to reach this conclusion.

This section has been reworded and cut in order to align with the removal of this method.

**Line 758 Figure 11: Remove erosion/deposition scale bar from figure since apparently not used. In caption, explain how vegetation stability was assessed.**

This figure has now been replaced due to the increased complexity of representing both vegetation influence and morphological change spatially, and instead with the inclusion of new excess drag calculations has been replaced with a more functional group specific analysis of morphological change compared to inundation depths. It is hoped that this will improve on the previous figure by showing spatial relation to the channel, vegetation type, and the erosional

signal located here for a specific time period and flood event, as opposed to the broader groupings in the previous figure.

Lines 765-766: "...there is clear evidence of light green patches where dark green patches may be expected had the vegetations [sic – should be vegetation's] stabilizing effect not been present." This is an intriguing idea, but no details are provided for why dark green patches might be expected in these regions. Explain why this is reasonable.

As above this section has been removed.

Line 783ff: Acquiring and processing UAV or TLS data represents a major investment in equipment and technician training. The presented data set (multiple repeat flyovers with different techniques) is much more detailed than what would be available for most (all?) other sites. Therefore, it would be extremely helpful to have authors leverage the current data to assess best practices and minimum collection needs that should be acquired in other settings. For example, if only one UAV overpass were possible, at what time of year should be it be flown, and using which technique (LS, MS, or RGB)? Would the answer depend on the type of vegetation characterized and, if so, how? This is a huge missed opportunity for this data set.

This is an excellent point, and one that has been incorporated in to the discussion of what is next for traits based methods to involve when is the best time to capture data, and how applicable are the methods (*Lines 944-951*). Having such a complete dataset makes it useful for discerning what is the most useful method and when is best to capture data, but such a comparison is currently beyond the scope of the current paper and is one for revisiting in future, especially given the current length of the paper. Likewise, the datasets collected in this study are available open access within the paper for others to use and research with, and we hope that is the case.

**Technical Comments**

The text is generally well written and clear, though there are several glaring exceptions.

**Line 180 (and elsewhere in text): The use of nested parentheses is odd and confusing. Considering using a semi-colon within a single set of parentheses.**

Line 184:

Changed accordingly so the reference is not in parentheses.

**Line 207ff: "spatial and temporal (i.e., 4D) variation" plus later "planform evolution": this reads like 2D + time = 3D measurements. The text needs to clearly explain/argue how it is truly 4D.**

**Lines 218-228:**

This is resolved through the removal of the longer term planform section which was causing confusion, and an emphasis on why the work is presented as 4D. This first aim has now been removed, with the latter aims expanded upon.

**Line 218: The word "exemplar" suggests that the study site is somehow better than other sites. Either explain in text why it is so outstanding and uniquely qualified for this type of study, or (if you instead want to suggest that the same methods could work elsewhere) then use a different term.**

This term was originally used not to suggest it was better than other sites, but based on the definition that it was a typical or good site, and such was appropriate to study this method. To avoid confusion in interpretations of the words meanings, we have removed this from the text.

**Line 226: The phrase "starting from the earliest gauge record" is confusing and unclear. Rewrite. Would also be nice to specify that the gauge period of record was from 2002 to 2021.**

This has been removed from the text to avoid any confusion, with the caption providing information on the source and length of data availability in line with comments from Reviewer 1.

**Line 229 Figure 1: Label each subpanel (there are at least 4) with a letter for easy reference in text. In the map, delineate the area used for the decadal analysis and shown in Figure 4A. Increase font size of all text in the discharge plot. Make sure that the exceedance level for 1.48 m in legend is consistent with text (is it 99% or 99.9% exceedance?).**

The figure has been adjusted accordingly, but has been labelled with three rather than four panels as it seems unnecessary to reference both of the context maps as separate panels when there role is the same. The original figure was for 99% flow exceedance and so has again been adjusted to reflect this.

**Line 241 (and elsewhere in text): Incorrect punctuation (semi-colon). Fix.**

Error noted, text removed and checked elsewhere.

Line 270: "To investigate the morphological process of avulsions..." The rest of this paragraph has a totally different topic than the first two sentences; move to a separate paragraph. In addition, this section discusses UAV flood imagery, which has not yet been presented; it would be helpful to move this section until after the UAV images have been discussed.

Removed from the text in line with the removal of this analysis.

**Line 277: It appears from Table 1 that one TLS survey was used. Change "TLS surveys" to be singular.**

Originally this was due to the description of multiple scans at the site on the same day, but to be in line with the singular in the table this has been changed.

- Line 296: Spell out abbreviation GNSS.
- Line 298: Spell out abbreviation GCP
- Line 301, 318, etc.: Make sure that the entire methods section is in the past tense to describe what you did.

In line with suggestions from Reviewer 1, this section has been removed and the reader prompted to refer to the methods paper that accompanies the work. As such, these abbreviations are not defined.

**Line 391: Comma splice. Fix.**

Removed.

**Line 464 (and perhaps elsewhere): Avoid contractions in formal writing.**

Noted, this has been removed and no other contractions where found.

**Lines 513-514: "...vegetation being modelling." Fix grammar.**

Line 489:

Changed to modelled in line with reviewer 1 comments also.

**Line 563 Figure 7: Increase all font sizes, especially of x- and y-axis values, to at least 10 point font.**

Figure has been changed to increase the font point sizes to make them more legible as suggested.

**Line 765: Eliminate second person ("you") from document.**

The section of text in which the second person you has been removed, and the remainder of the manuscript checked to ensure this has not been replicated elsewhere.

Line 839: Spelling of "geomprohic."

Corrected.

Line 880ff: The reference list (and in-text citations) should use a consistent formatting. The Kattge et al. paper unexpectedly includes "and" between each author, and also arguably could have its author list curtailed. In O'hare et al. 2011, the "H" of M. O'Hare is not capitalized, whereas the same author's H is capitalized in O'Hare et al. 2016, which is in the same journal.

This appeared to be in the case of the Kattge paper a result of the reference manager style, as a result it has been edited to reduce the number of authors shown and show the last author.

The O'Hare references have been fixed in the reference manager, but also appeared to have an anomaly in the styles manager, whereby the letters after O' were not being capitalised, as this was also found to be the case for O'Briain. This has been amended so all references are correctly capitalised.

We hope that the editors and reviewers are happy that we have fully addressed all comments and are happy to clarify any responses and changes as required. Thank you for considering our manuscript for publication in Earth Surface Dynamics.

MAG

Dr Christopher Tomsett School of Geography and Environmental Science University of Southampton

---

## Author Response (AR2)

Dr Christopher Tomsett
School of Geography and Environmental Science
University of Southampton
Southampton
SO17 1BJ

26th June 2023

**ESURF-2021-102: Exploring the 4D scales of eco-geomorphic interactions along a river corridor using repeat UAV Laser Scanning (UAV-LS), multispectral imagery, and a functional traits framework.**

**Response to Comments from Reviewers**

Dear Lina,

We thank you and the reviewer for your constructive and helpful comments on our submitted manuscript. Below we outline how we have addressed the reviewers overall feedback and the individual comments. For clarity, we summarise the review comments in **_bold and italics_** and list our response below each one:

**Reviewer 1:**

***The revised manuscript is a tighter, more focused contribution to the understanding of the coupled evolution of river-corridor vegetation and geomorphology. The document is now exclusively focused on the two-year period when detailed high-resolution surveys were performed, as well as the ~1 km reach where they were obtained, thus providing an opportunity to examine vegetation-sediment feedbacks in real time.***

***Unfortunately, however, this revised focus brings into sharp clarity several underlying weaknesses. The manuscript describes itself as an "exploratory analysis" on Line 889, a term that succinctly confirms that an intriguing sequence of analytical steps was performed, but that the limitations and uncertainties of the chosen method(s) were poorly characterized, and as such the outcomes are challenging to interpret and use to motivate further work.***

***First, there remains no field validation of remote sensing metrics or guild assignments. The authors summarize in the Introduction the importance of direct measurement of guild characteristics (frontal area, flexibility, etc.), as well as how different individuals from the same species may have different growth forms at different sites, yet in the end they rely on remote sensing algorithms for these determinations, without any field validation at this site for these individuals. Classification results (which require a substantial amount***

*of manual attention and parameter choices) are compared to previous findings (TRY database) for these species at other locations; this is better than nothing, but (as noted above) remains an imperfect comparison. Without any field validation, all trait assessment should be considered preliminary or suggestive, rather than quantitative and authoritative. As such, it is difficult to assess whether the primary negative results in the paper (that there was no consistent pattern of erosion or deposition within different functional groups) results from lack of signal, or lack of ability to detect an actual signal.*

The initial scope of our field campaigns was not to target traits based methods specifically, but vegetation interactions more generally, it is regretful that no specific traits based data was collected in the field for validation. This is something that with hindsight we would certainly do should we attempt to replicate the research elsewhere, and as such is commented on in section 6 of the paper.

As the reviewer correctly points out, the extraction of these traits is predominantly determined by remote sensing methodologies as explained within our methods. However, to add some additional estimation of the error associated with the extracted values (beyond the previous revisions which compared them to wider literature) an assessment of both the UAV-LS and QSM methods has been undertaken and included in the manuscript.

The former compares the DBH and tree height values of trees that are within both the TLS scans and the two closest UAV-LS scans temporally before and after this survey date. This enables us to establish how well the UAV-LS methods performed in comparison to TLS which is widely regarded as the benchmark in high accuracy remote sensing.

In addition, to assess the outcomes of the QSM methods, manual measurements of tree height and DBH for all four remotely sensed groups were performed on the raw TLS and UAV-LS point clouds in CloudCompare, and these were then compared to the resultant values from the QSM methods.

These datasets are then used to assess bias or variation occurring during the trait extraction process (new Figure 4). Overall, this analysis allowed us to quantify the uncertainty in the vegetation metric extraction methods and provided some additional confidence that the methods used are reconstructing vegetation well. We hope that in the absence of manually measured field validation data, the use of TLS data and this additional analysis provides some further measure of the quality of the plant characterisation techniques for the reader,

*Second, numerical (Delft3D) modeling was introduced to sharpen the focus on the reach-scale implications of the various vegetation characteristics, but there was apparently no field validation of the numerical model. The surveyed flood extents during Storm Dennis provided the minimum initial/boundary conditions to run the model; there were apparently no separate measurement(s) of water depth, discharge, or velocity within the model domain that could be used to validate the numerical representations of functional group drag (Lines 461-485). Without validation, the modeling "results" (e.g., Figure 11, Table 3) become preliminary or suggestive and should not be treated as robust outcomes.*

The introduction of a numerical model was made in the last set of revisions in an attempt to address the comments of reviewers in relation to interpretation of the results, and of course no

in field measurements had been taken during the flood conditions during the field campaign. It was only ever supposed to be indicative, to explore the ways in which depth dependency may be influencing morphological change. We agree that the way this exploratory analysis was presneted perhaps overstated the output from the model and we have rewritten the discussion around this to tone down the interpretations. In this vein, the annual approach and analysis has been removed from the manuscript as this confuses the simple core message that we are trying to convey.

We retain an example of morphological change over one winter based on the high flow output, but have moved the presentation of this from the results section to the discussion, to help emphasise that this is exploratory analysis undertaken for indicative hypothetical flows based loosely on observed data. Taken together, these changes shift the focus from this section being treated as 'results' to them being considered 'preliminary' and 'suggestive' as recommended by the reviewer.

***Third, the article title ("Exploring the 4D scales of eco-geomorphic interactions…) remains imperfectly matched to its contents. Although the 3D point cloud data are used for vegetation assessment, the land cover assessments (Table 2, Figure 6) remain 2D, with only one vegetation type assigned to each horizontal pixel; understory and overstory plants are not distinguished. Without 3D spatial analysis, this is not a 4D analysis. In addition, the analysis is only completed at one spatial scale (individual plant --> convex hulls of unspecified size but presumably a few meters across --> pixels at reach scale), so multiple scales are not explored. As the authors point out in the Introduction (Lines 35-49), scaling from individual plants to >1 km reaches remains a challenge, but not enough information is provided to conclusively claim that they succeeded here. The current analysis apparently did not even include a training (calibration) and a test (validation) data set (Lines 419-420) to assess efforts.***

We have changed the title of the paper to better represent the content, accepting that the '4D' assertion is not fully justified.

We do believe that vegetation analysis has been undertaken across scales as identified in the reviewer comment, occurring at both the individual plant and reach scale during different parts of the analysis. However, in line with the suggestions we have also reduced the focus on scales explicitly, only retaining reference to this when explaining the methods (in response to the 5[th] comment below).

With regard to the training and calibration dataset, the decision not to split this data was based on the number of samples collected, and the fact that increasing this significantly was not an option due to the time intensive nature of manual selection and analysis. To have split a few samples off would have left too small a dataset to undertake meaningful statistical analysis. At the same time, it would have also reduced the number of samples to train the model on. Likewise, in the lines that follow 419-420, the method of random forest classification undertakes out-of-bag accuracy, the method by which subsamples are tested for each decision tree. This can be used to infer a primary level of model accuracy, and is one used widely in the literature. We have supplemented this with a test dataset from high resolution ortho-imagery and field notes to check predicted accuracy, which is detailed in sections 3.2.5. and presented in 4.2.4. and Figure 8.

***Fourth, for a manuscript that focuses on remote sensing, it remains extremely odd that the Introduction does not summarize lessons learned and knowledge gaps from remote sensing of riparian vegetation.***

In line with changes to the introduction and background sections that have aimed to streamline the paper, more emphasis has been placed on including some of the wider remote sensing of vegetation literature, and showing how a knowledge gap exists in relation to using these methods to identify traits at greater spatial scales. Given that the manuscript is already lengthy, we have refrained from including an extensive review of the literature, instead pointing to some important review papers in addition to the changes outlined above.

***Fifth, even though the remaining contribution of this paper is its innovative methods, numerous key analytical details remain unspecified. Although it is nice that the flight dates are clearly laid out (Table 1), it appears that results were combined into only two vegetation maps (Line 405 and Figure 6). This combination should be clear from the beginning (i.e., Table 1 and Lines 248ff). In addition, justify why wintertime data were combined with subsequent summer data, when it appears from Figure 1c that many high flows occur during late winter/spring so therefore winter vegetation may not be comparable to vegetation the following summer. Some of the classifications rely on TLS data, which were only collected once. It is unspecified/unclear when/where several of the steps (point cloud segmentation, trait metric extraction) were performed, and whether this occurred for repeat scans or not. (For example, were all 24 individual tree segmentations performed from data collected on the same date, or multiple different dates?) It is unfortunate that the authors wrote in their response to comments that they consider this level of detail difficult to provide, since it remains impossible to understand, duplicate, or apply these methods without understanding what was done here.***

The methodological approach we used was complicated and it is clear that the overall workflow was not communicated clearly enough within the manuscript. Due to the (not always linear) methodological workflow undertaken to go from the initial field data to final extracted metrics, it is challenging to convey how each section links together, especially when in addition the processing is sometimes performed on different datasets at different scales.

We have attempted to improve the clarity around the methods in three ways in the revised manuscript:

1) Introduction of a new workflow figure to help readers understand which surveys were used for each different section of the methods (new Figure 2), and also to show how each of these processes work together to progress from raw data to the final extracted values. The flow diagram also includes reference to numbered sections and figures in the manuscript to help the reader navigate through when these processes occur.

2) We have carefully worked through the manuscript and introduced explicit use of the terms plant, group and reach scale, to make it clear whether the analysis is undertaken on a plant level basis, or across the reach. This also helps when referencing methods in the discussion. Plant and reach scale data have also been included in the flow diagram referenced above to help better direct the reader.

3) The classification approach that we used requires both leaf on and leaf off data, and is justified due to the analysis of reach scale characteristics of each group. This meant that we necessarily needed to combine winter and following summer data. We have commented on this point in both the methods (to justify the decision) and future work section (to discuss it as a limitation). We have also tried to better present and explain the overlap in the timings of different surveys, and how they feed into data collection and analysis. Specifically, in the methods section, we now refer to the month and year of each survey and, where required, point the reader to the survey (Table 1) to which we are referring.

*Sixth, the authors' response to comments repeatedly mentions the importance of conciseness, yet the authors' attention to this important consideration seems inconsistent. There are numerous places where excess information is included (e.g., overly authoritative discussion of bank erosion on Lines 66-74, overly loquacious discussion of speculative findings on Lines 649-729, overly optimistic Discussion on Lines 730ff), while the authors omit key information that describes what they did or justifies why it makes analytical sense.*

In revising the manuscript we have taken on board the reviewers comment that parts of the text are extraneous and have made extensive edits (mainly through reduction and removal of text) to address this. We hope that the changes outlined above to the methods section, and edits to the results and discussion sections, have helped to improve the manuscript by making it clearer what was undertaken when, why it was done, and the resultant outputs. We have also reduced the length of the introduction and discussion sections in order to compensate for the addition of the extra methodological detail requested. This has led to an overall reduction in paper length, despite an increase in the length of the methods.

*A few remaining specific comments:*

*Line 232: The site description states there are "two distinct reaches," which are not shown on the map, not obvious from the map, and apparently not discussed anywhere else in the manuscript. Elsewhere, the word "reach" seems to be used consistently to refer to the entire study area. It is recommended that different terms be used to refer to the upper and lower portions of the study reach.*

This was an oversight in the transition from the original manuscript which contained the long term analysis of the study site, where the upper and lower reaches were referred to more explicitly. As suggested, the term reach is altered to refer to the entire study area in line with the rest of the manuscript, and the upper and lower portions of the reach referred to as sections.

*Line 581: Unclear how the "over classification of shrubs" was assessed, since there was no independent validation of the ortho-imagery. In addition, unclear where this overclassification occurs on the landscape (Figure 6).*

Attempts to clarify this statement have been made in reference to the spatial pattern whereby areas classed as trees consistently have surrounding classification outputs shown as being

shrubs. This is especially the case for isolated trees which have small classification of shrubs, and are probably the result of image segmentation cutting out lower portions of the tree structure more so than a small area of shrubs being present. However, it is acknowledged that use of the terminology 'over-classification', rather than a more qualitative description as is now presented, did not well present the narrative that we were trying to convey.

***Line 613: It is claimed that "single branching herbs" were misclassified as "branching herbs." This is a confusing statement. Be consistent with functional group names throughout the text, and perhaps avoid two names that are this similar.***

We acknowledge the confusion caused here which is due to incorrect inclusion of the term branching in single stemmed herbs. The naming of these two groups has been checked throughout the manuscript to avoid any inclusion of the term branching when referring to single stemmed herb groups. The figures have also been updated to increase clarity in reference to each of the individual groups.

***Finally, the manuscript is plagued by numerous punctuation mistakes (e.g., Lines 173, 221, 288) as well as spelling issues (e.g., Lines 239, 584, 649, 666, 955, 960, 961). In addition, it is very difficult to read the text on some axis labels (e.g., Figures 8 and 9).***

The specific mistakes in the text highlighted above have been corrected, as well as several others found whilst reviewing the manuscript. Text in figures 8 and 9 (now 9 and 10), has also been increased in size for histogram subsets, and for figure 9 (now 10) we have also attempted to reduce the amount of information present in the figure by removing repetition of phrases such as net volume change and the volume change axis labels to only appear once.
* * *
We hope that the editors and reviewers are happy that we have fully addressed all comments and are happy to clarify any responses and changes as required. Thank you for considering our manuscript for publication in Earth Surface Dynamics.

Dr Christopher Tomsett
School of Geography and Environmental Science
University of Southampton

---

## Author Response (AR3)

Dr Christopher Tomsett
School of Geography and Environmental Science
University of Southampton
Southampton
SO17 1BJ

**20/10/2023**

**ESURF-2021-102: Using repeat UAV based laser scanning and multispectral imagery to explore eco-geomorphic feedbacks along a river corridor. (*Formerly: Exploring the 4D scales of eco-geomorphic interactions along a river corridor using repeat UAV Laser Scanning (UAV-LS), multispectral imagery, and a functional traits framework*).**

**Response to Comments from the Associate Editor**

Dear Lina,

We thank the Earth Surface Dynamics editors for their constructive and helpful comments on our submitted manuscript. Below we outline how we have addressed the comments from yourself. For clarity, we summarise the review comments in **bold and italics** and list our response below each one, we also provide a list of other changes below:
* * *
**Section 4.2 of Results**

**Names of trees should not be capitalized, e.g., 'alder', 'willow', etc…**

Names of vegetation have been changed to remove capitals.

**Also, be sure to provide latin names of species studies (e.g., Alnus incana?) at their first mention**

Latin names for species identified and used for traits comparisons included in the text.

**Section 4.2.4:**

**If channels are commonly re-activated during winter, I wouldn't call them paleochannels, but perhaps side channels.**

We agree with this new terminology and have changed accordingly.

**School of Geography and Environmental Science**
University of Southampton, Highfield Campus, Southampton SO17 1BJ United Kingdom
Tel: +44 (0)23 80592285  Fax: +44 (0)23 80593295  www.southampton.ac.uk/geography

**"over classification" should be one word or separated with a hyphen (similar to overestimate used elsewhere in text)**

Changed to overclassification inline with the format used for overestimate, both in main body text and figure captions.

**Section 5.2**

**Change to 'this presents a limitation for some research.'**

Changed.

**Section 6**

**Change to 'an how these representations change temporally'**

Changed sentence structure accordingly.

**Next line; change to : a plant's life**

Changed.

**Change to: 'their complexity due to the sensors' limits of detection'**

Changed.

**Here, and throughout text: make sure that 'ground-based' has a hyphen if used as an adjective**

Changed in all instances.
* * *
**Other Changes made by the Authors:**

Removal of hyphen in reach-scale from first instance in abstract to be consistent with remainder of manuscript.

Missing punctuation at the end of line 53 corrected.

Lines 191 to 195 altered, correcting some changes in tense and including the period of time used for the river depth statistics.

Caption for Figure 1 changed, indicating source material for the country outlines background mapping in inset A (EUROSTAT, 2020) and inset B (UK Office for National Statistics, 2019), as requested by file validation process.

Line 210 added in UAV-RGB to sentence when specifying GSD or imagery.

Line 391, clarification that 300 forests were used in the random forest model.

Sections 4.2.X renamed to 4.1.X after error in number formatting.

Section 4.1.3. and 4.1.4., inclusion of vertical prior to 'skew' and 'skewness' to make clear this is vertical variation.
* * *
We hope that the editors and reviewers are happy that we have fully addressed all comments and are happy to clarify any responses and changes as required. Thank you for considering our manuscript for publication in Earth Surface Dynamics.

Dr Christopher Tomsett
School of Geography and Environmental Science
University of Southampton